# Histone H3G34R mutation causes replication stress, homologous recombination defects and genomic instability in *S. pombe*

Rajesh K Yadav[1], Carolyn M Jablonowski[1], Alfonso G Fernandez[1], Brandon R Lowe[1], Ryan A Henry[2], David Finkelstein[3], Kevin J Barnum[4], Alison L Pidoux[5], Yin-Ming Kuo[2], Jie Huang[6], Matthew J O'Connell[4], Andrew J Andrews[2], Arzu Onar-Thomas[6], Robin C Allshire[5], Janet F Partridge[1]*

[1]Department of Pathology, St. Jude Children's Research Hospital, Memphis, United States; [2]Department of Cancer Biology, Fox Chase Cancer Center, Philadelphia, United States; [3]Department of Bioinformatics, St. Jude Children's Research Hospital, Memphis, United States; [4]Department of Oncological Sciences and Graduate School of Biomedical Sciences, Icahn School of Medicine at Mount Sinai, New York, United States; [5]Wellcome Trust School for Biological Sciences, University of Edinburgh, Edinburgh, Scotland; [6]Department of Biostatistics, St. Jude Children's Research Hospital, Memphis, United States

**Abstract** Recurrent somatic mutations of *H3F3A* in aggressive pediatric high-grade gliomas generate K27M or G34R/V mutant histone H3.3. H3.3-G34R/V mutants are common in tumors with mutations in p53 and ATRX, an H3.3-specific chromatin remodeler. To gain insight into the role of H3-G34R, we generated fission yeast that express only the mutant histone H3. H3-G34R specifically reduces H3K36 tri-methylation and H3K36 acetylation, and mutants show partial transcriptional overlap with *set2* deletions. H3-G34R mutants exhibit genomic instability and increased replication stress, including slowed replication fork restart, although DNA replication checkpoints are functional. H3-G34R mutants are defective for DNA damage repair by homologous recombination (HR), and have altered HR protein dynamics in both damaged and untreated cells. These data suggest H3-G34R slows resolution of HR-mediated repair and that unresolved replication intermediates impair chromosome segregation. This analysis of H3-G34R mutant fission yeast provides mechanistic insight into how G34R mutation may promote genomic instability in glioma.

*For correspondence: janet.partridge@stjude.org

Competing interests: The authors declare that no competing interests exist.

## Introduction

The genomic DNA of eukaryotes is packaged into chromatin, which regulates all DNA transactions including transcription, replication and repair. The basic subunit of chromatin is the nucleosome, which is made up of 147 bp of DNA wrapped around a core octamer of histone proteins. Proteins that regulate chromatin dynamics are frequently mutated in cancer, and recently mutations in the histone genes themselves have been discovered (*Schwartzentruber et al., 2012*; *Wu et al., 2012*; *Behjati et al., 2013*). One such mutant is the Glycine 34 to Arginine (G34R) mutant of histone H3.3, identified as a frequent somatic mutation in pediatric high-grade cortical glioma (pHGG). H3.3 is a non-canonical H3 variant, whose deposition is not linked to replication and which can accumulate in post-mitotic cells and in areas of high transcriptional activity (*Skene and Henikoff, 2013*). Notably, G34R mutations are only found in H3.3, primarily in one of two H3.3 genes (*H3F3A*) and not in the

**eLife digest** Children suffering from a brain cancer called high-grade glioma rarely recover because there are no therapies that can effectively target this disease. Recently, mutations in a gene that encodes a protein called histone H3 were found in children's glioma cells. Histone proteins bind to DNA to help package it into cells. They help to protect the DNA from damage, control when genes are switched on or off, and influence how the DNA is copied when cells are preparing to divide to produce two daughter cells. However, little was known about why one of the histone H3 mutations is associated with high-grade gliomas.

Humans and other animals have many different versions of the histone H3 protein, which makes it difficult to study a mutation that only affects one particular version. Therefore Yadav et al. used fission yeast – which they engineered to only contain one form of histone H3 – to study what effect the mutation has on cells.

The experiments show that fission yeast cells with the mutant form of histone H3 took longer to replicate their DNA and were more likely to die when exposed to chemicals that interfere with DNA duplication. Furthermore, the mutant cells were slower at repairing certain types of damage to DNA and were more likely to go on to divide before DNA duplication and repair were completed. As a result, the daughter cells produced were more likely to have gained or lost whole chunks of their DNA. This phenomenon is known as chromosomal instability and is often seen in cases of high-grade glioma in children.

The findings of Yadav et al. give new insight into how a specific mutation affecting the histone H3 protein may act in cells. Future experiments will need to confirm whether this mutation also has a similar effect on children's glioma cells, which may help to provide new options for treating this cancer.

replication coupled H3 proteins H3.1 and H3.2, which are encoded by a total of 13 genes. G34R mutations are commonly found in tumors additionally mutant for the tumor suppressor p53 and ATRX, a chromatin remodeler involved specifically in nucleosomal deposition of the H3.3 variant histone at telomeres and pericentric heterochromatin (*Goldberg et al., 2010*; *Lewis et al., 2010*; *Drané et al., 2010*). While p53 and ATRX mutations may contribute to the genomic instability exemplified by these gliomas, the specific role of the G34R H3.3 mutant in disease remains a mystery.

One published study showed G34R nucleosomes had reduced K36me2/3 on the same H3.3 tail (*Lewis et al., 2013*), suggesting a potential role in regulating H3K36 methylation in gliomas. Furthermore, mutations in SETD2, the enzyme that performs H3K36 di- to tri-methylation in mammals (*Edmunds et al., 2008*), have also frequently been identified in pHGG (*Fontebasso et al., 2013*). However, SETD2 mutations are not found in conjunction with histone mutations, suggesting the importance of disruption of the H3K36me3/ SETD2 axis in cortical pHGG. To gain biological insight into the role of the G34R mutant, we engineered fission yeast to express either wild-type or G34R mutant histone H3. In fission yeast, a single enzyme, Set2, performs all methylation on H3K36 (*Morris et al., 2005*). We hypothesized that H3-G34R mutation may reduce Set2 function and influence important chromatin-templated processes such as DNA transcription, replication and repair, which are highly conserved between fission yeast and human.

Here, we show that histone H3-G34R mutation specifically reduced tri-methylation, but not di-methylation of H3K36, and caused a decrease in H3K36 acetylation. However, G34R mutants displayed different phenotypes to *set2Δ*, suggesting that defective H3K36 modification may not be the sole defect in G34R. H3-G34R mutants showed chromosome instability and sensitivities to DNA damaging agents which are distinct from cells lacking Set2. H3-G34R mutants had defects in HR-directed repair, which may be attributed to a delay in DNA repair dynamics at compromised replication forks. Together, our work provides valuable insight to the potential role of H3-G34R mutations in pediatric high-grade gliomas.

## Results

In contrast to the complexity of mammals, where fifteen genes encode three different histone H3 proteins (H3.1, H3.2, H3.3), fission yeast have three genes that code for a single histone H3 protein (*Figure 1a*) (*Matsumoto and Yanagida, 1985*). Fission yeast H3 has features of both H3.1 and H3.3 proteins of higher eukaryotes (*Figure 1—figure supplement 1a*). To simplify the analysis of histone H3 and H4 mutants, strains have been derived that express only one H3 and one H4 gene (*hht2*+ and *hhf2*+) (*Mellone et al., 2003*). These strains maintain histone protein levels similar to those in wild-type strains (*Figure 1—figure supplement 1b*), likely through upregulation of *hht2*+ and *hhf2*+ expression. To assess the effect of G34R mutation, we introduced mutations into *hht2*+ (*Figure 1a*), and derived strains that express only the mutant histone H3 gene (*Figure 1—figure supplement 1c*). These strains are viable at a range of temperatures and appear to have normal chromosomal integrity (*Figure 1—figure supplement 1d,e*). 'Sole copy' histone H3 and H4 strains were used throughout this study and are named **H3-WT** and **H3-G34R** in the text, and '**WT**' and '**G34R**' in the figures. Additional mutants were all in the sole copy H3-WT background unless indicated otherwise by inclusion of (3xH3) denoting 3 copies of H3/H4.

### Post translational modification of H3K36 is altered in H3-G34R mutants

In mammalian cells, ectopic expression of H3.3G34R reduced K36me2/3 methylation on the mutant histone tail (*Lewis et al., 2013*). Five enzymes can methylate H3K36 in mammals, with only SETD2 responsible for H3K36me3 (*Edmunds et al., 2008*). In fission yeast, a single enzyme (Set2) performs all methylation on H3K36 (*Morris et al., 2005*). We began our investigation by determining if H3-G34R mutation reduces Set2 H3K36 methyltransferase activity. Because of the proximity of G34R to K36, we tested whether antibodies can specifically recognize K36 methylation states in the G34R mutant. Using di- or tri-methylated K36 H3 peptides as targets, we identified two anti-K36me3 antibodies that were minimally affected by G34R mutation, and one antibody against K36me2 whose binding was weakened ~10 fold by the G34R mutation (*Figure 1b*, *Figure 1—figure supplement 1f*). *Figure 1—figure supplement 1f–g* serve to demonstrate reproducibility of the H3K36me3 methylation pattern with an alternate H3K36me3-specific antibody. Using these antibodies, we found that H3K36me3 was markedly reduced in H3-G34R compared to H3-WT cell extracts, but that H3K36me2 levels were unchanged (*Figure 1c*, *Figure 1—figure supplement 1g*). As we observed that binding of the H3K36me2 antibody is reduced ~10 fold on G34R peptides, these results suggest that K36me2 is elevated in H3-G34R cells. Thus Set2 function is altered in H3-G34R cells, with specific reduction of H3K36me3 and retention/ accumulation of H3K36me2.

Since endogenously tagged Set2-FLAG protein levels were similar in H3-WT and H3-G34R cells (*Figure 1—figure supplement 1h*), the reduction in H3K36me3 is not caused by a loss of *set2*+ expression in H3-G34R cells. Additionally, chromatin immunoprecipitation (ChIP) analysis revealed that Set2-FLAG recruitment to chromatin was not reduced in H3-G34R cells (*Figure 1d*), suggesting that the H3-G34R mutation hinders Set2 activity but not access to K36. We therefore asked if we could force the generation of H3K36me3 in H3-G34R cells by overexpression of the Set2 methyltransferase (*Figure 1—figure supplement 1i*). Western analysis of extracts prepared from cells that overexpress Set2 showed that pSet2-FLAG restored K36me2 and me3 in H3-WT *set2Δ* strains but was able to generate only low levels of H3K36me3 in both H3-G34R and H3-G34R *set2Δ* cells (*Figure 1e*). Thus the H3-G34R mutation reduces Set2 activity on the H3 tail, leading to a specific reduction in tri-methylation of H3K36, and this effect cannot be bypassed by overexpression of Set2.

In addition to methylation of H3K36, this residue is also regulated by acetylation (*Morris et al., 2007*). To address whether H3-G34R impedes acetylation at H3K36, we used quantitative mass spectrometry to perform targeted analysis of acetylation of tails of histone H3 and H4 in histones acid extracted from H3-WT, H3-G34R and *set2Δ* strains (*Figure 1f*) (*Kuo and Andrews, 2013*). Calibration was performed to monitor the elution time of the acetylated and propionylated tryptic peptides and transitions were created to study acetylation of H3-WT and H3-G34R as well as the H4 tails (see *Supplementary files 1* and *2* and Materials and methods). These studies showed acetylation of K36 on histone H3 was greatly reduced in H3-G34R strains and somewhat reduced in *set2Δ*. Additionally, there was a slight upregulation of H3K18 acetylation and possibly enhanced K27 acetylation in H3-G34R, although K27 acetylation in H3-WT was highly variable. *set2Δ* cells showed more widespread changes, with a slight reduction of H4 K16 acetylation, some induction of H3K9 and K18 acetylation

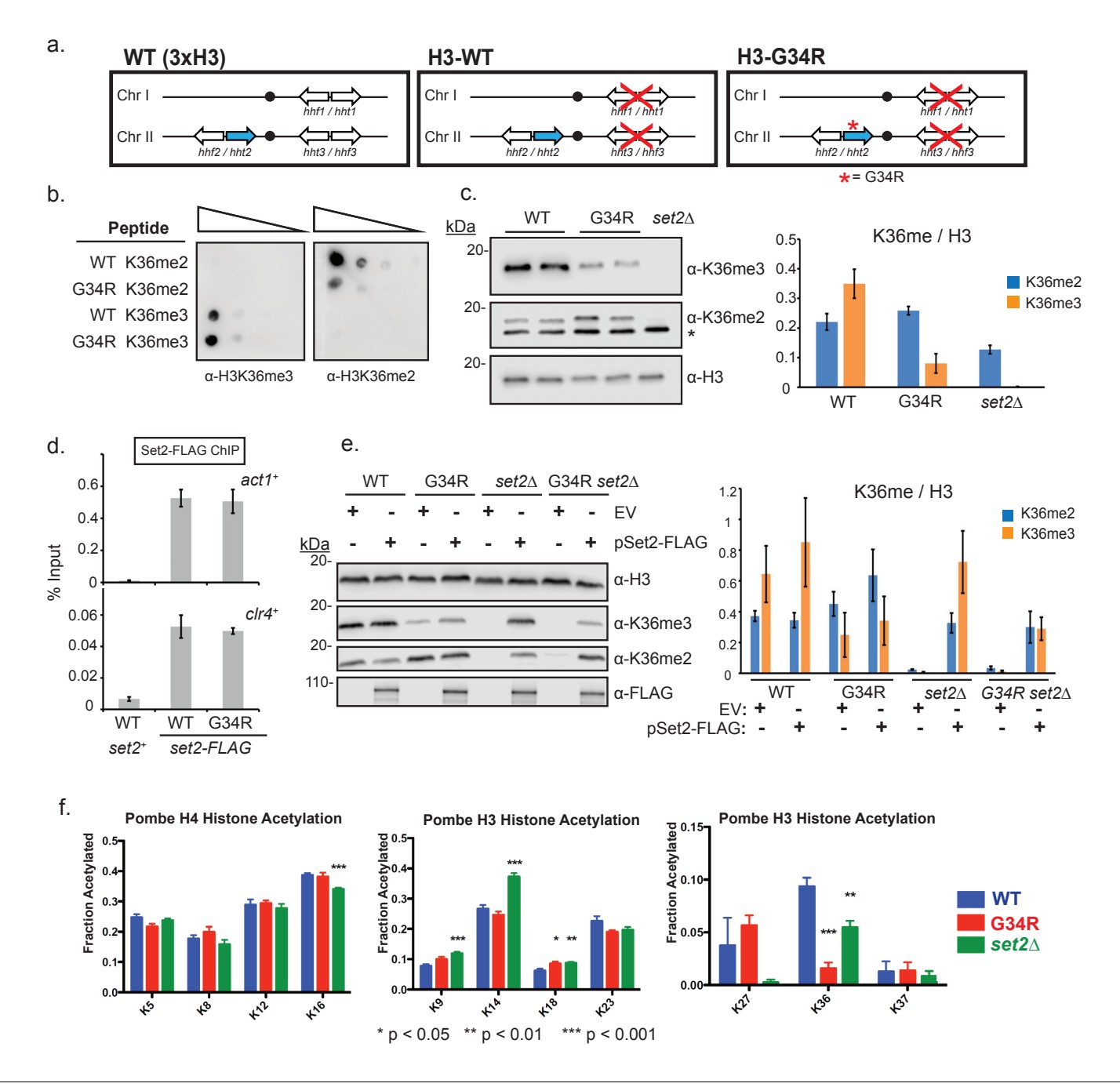

**Figure 1.** Post translational modification of H3K36 is altered in H3-G34R mutants. (**a**) Scheme of the histone H3 (*hht*) and histone H4 (*hhf*) genes in *S. pombe* highlighting the H3 gene (*hht2*) in which mutations were engineered (blue), and representation of the H3-WT and H3-G34R strains used in this study. (**b**) Dot blot analysis to quantitatively assess whether Anti-K36 methyl antibodies equivalently recognize WT and G34R peptides with different K36 methyl modifications. (**c**) Western blot analysis of K36me2, K36me3, and total H3 in H3-WT, H3-G34R and *set2Δ* cell extracts. Star marks nonspecific band. Replicate westerns were run for K36me2 and K36me3 including 2 biological replicates for H3-WT and H3-G34R strains for quantitation. The mean results for K36me2 or 3 relative to total H3 are displayed on the right. (**d**) ChIP analysis of Set2-FLAG association with *act1+* and *clr4+* loci in H3-G34R and H3-WT cells, represented as % of input DNA. Data represent mean ± SEM from 4 experiments. (**e**) Western blot of K36me2 and K36me3 on overexpression of Set2-3xFLAG or empty vector (EV) in the indicated strains. Total H3 serves as a loading control. Quantitation of 3 sets of western blots from duplicate biological repetitions is shown on the right, with levels of methylated H3K36 normalized to total H3 protein. (**f**) Mass spectrometry of acetylated lysines in histone H4 and H3 tails in H3-WT, H3-G34R, and *set2Δ* cells. Data are averaged from triplicate analyses of 3 biological replicates. Please refer to *Figure 1—figure supplement 1*, and *Supplementary files 1* and *2* for additional information in support of *Figure 1*.

*Figure 1 continued on next page*

*Figure 1 continued*

The following figure supplements are available for figure 1:

**Figure supplement 1.** Characterization of the H3-G34R mutants.

**Figure supplement 2.** Characterization of cell cycle regulation of H3K36 methylation during the cell cycle.

and a marked increase in H3K14 acetylation. In summary, H3-G34R cells showed a focal pattern of chromatin changes centered on inefficient tri-methylation and reduced acetylation of histone H3K36, whereas *set2Δ* effects were more widespread.

H3K36me3 and H3K36 acetylation have been shown to be cell cycle regulated (*Li et al., 2013*; *Pai et al., 2014*), with K36me3 accumulating in G1 and early-mid S phase. We assessed the consequence of H3-G34R mutation on K36me3 accumulation at several stages of the cell cycle, using cells synchronized in G1, S, G2 and M phases (*Figure 1—figure supplement 2*). Although levels of H3K36me3 were reduced in H3-G34R cells compared to H3-WT at all stages of the cell cycle (*Figure 1—figure supplement 2a,b,c*), H3K36me3 did accumulate in both H3-WT and H3-G34R cells synchronized in G1, indicative that the G34R mutation does not override cell cycle control of Set2 function. Additionally, this experiment also demonstrates that H3K36me2 levels are higher throughout the cell cycle in H3-G34R compared to H3-WT cells.

## H3-G34R mutants are transcriptionally distinct from *set2* cells

Post-translational modifications of H3K36 are important for transcriptional control, raising the possibility of transcriptional disruption in H3-G34R cells. Cells that lack the H3K36 methyltransferase Set2 or that express a mutant of Set2 that allows only di-methylation of K36 show a similar widespread transcriptional upregulation in fission yeast (*Suzuki et al., 2016*; *Matsuda et al., 2015*), suggesting that H3K36me2 is not sufficient for silencing. Since H3-G34R cells exhibit reduced H3K36me3 and possibly enhanced K36me2, as well as a reduction in H3K36Ac, we asked how H3-G34R mutation influences transcriptional control.

In genome-wide RNA-Seq analyses, *set2Δ* had many transcripts with altered regulation (753 genes were up and 149 genes downregulated) (*Figure 2a,b*) in an organism with ~5100 protein coding genes and ~1500 non coding RNAs (*Supplementary file 3*). Unexpectedly, the H3-G34R mutant showed more mild transcriptional effects (repression of 69 genes and upregulation of 172 genes). This difference in gene regulation may stem from additional chromatin changes such as increased H3K14Ac in *set2Δ*.

143 genes were deregulated in both H3-G34R and *set2Δ* cells: 97 were coordinately regulated, with 9 genes downregulated, and 88 genes upregulated in both *set2Δ* and H3-G34R (*Figure 2b*). In most other cases (44/46), transcripts accumulated in *set2Δ* but were repressed in H3-G34R. As can be seen on chromosome-wide plots (*Figure 2c,d*), transcripts that are differentially regulated between H3-G34R and *set2Δ* map to regions adjacent to subtelomeric regions of chromosome 1 and 2 (ST or sub-subtelomeric domains [*Matsuda et al., 2015*; *Buchanan et al., 2009*]), which are not heterochromatic and are normally expressed at low levels (*Mata et al., 2002*). We confirmed gene regulation trends for 2 genes from ST domains by real time qPCR (*Figure 2e*), and additionally show that transcript levels for these genes in the double mutant, H3-G34R *set2Δ*, are similar to *set2Δ* cells. Thus H3K36me2 is perhaps critical for the repression in ST domains seen in H3-G34R cells. Intriguingly, these domains have been recently shown to associate with Shugoshin (Sgo2) to form highly condensed subtelomeric 'knob' regions which are transcriptionally silenced and delayed for replication (*Matsuda et al., 2015*; *Tashiro et al., 2016*).

Set2 has also been shown in both *S. cerevisiae* and *S. pombe* to repress antisense transcription (*Nicolas et al., 2007*; *Shim et al., 2012*; *Venkatesh et al., 2016*). Focusing on just these antisense transcripts, of which there are 635 in fission yeast, we found that, compared to H3-WT cells, 318 were differentially regulated in *set2Δ*, and 62 in H3-G34R mutants, with the vast majority being upregulated (313 and 57, respectively) (*Figure 2f*). 47 antisense transcripts were targeted by both Set2 and H3-G34R, and 43 of these accumulated in both mutants. Therefore H3-G34R cells, similar to

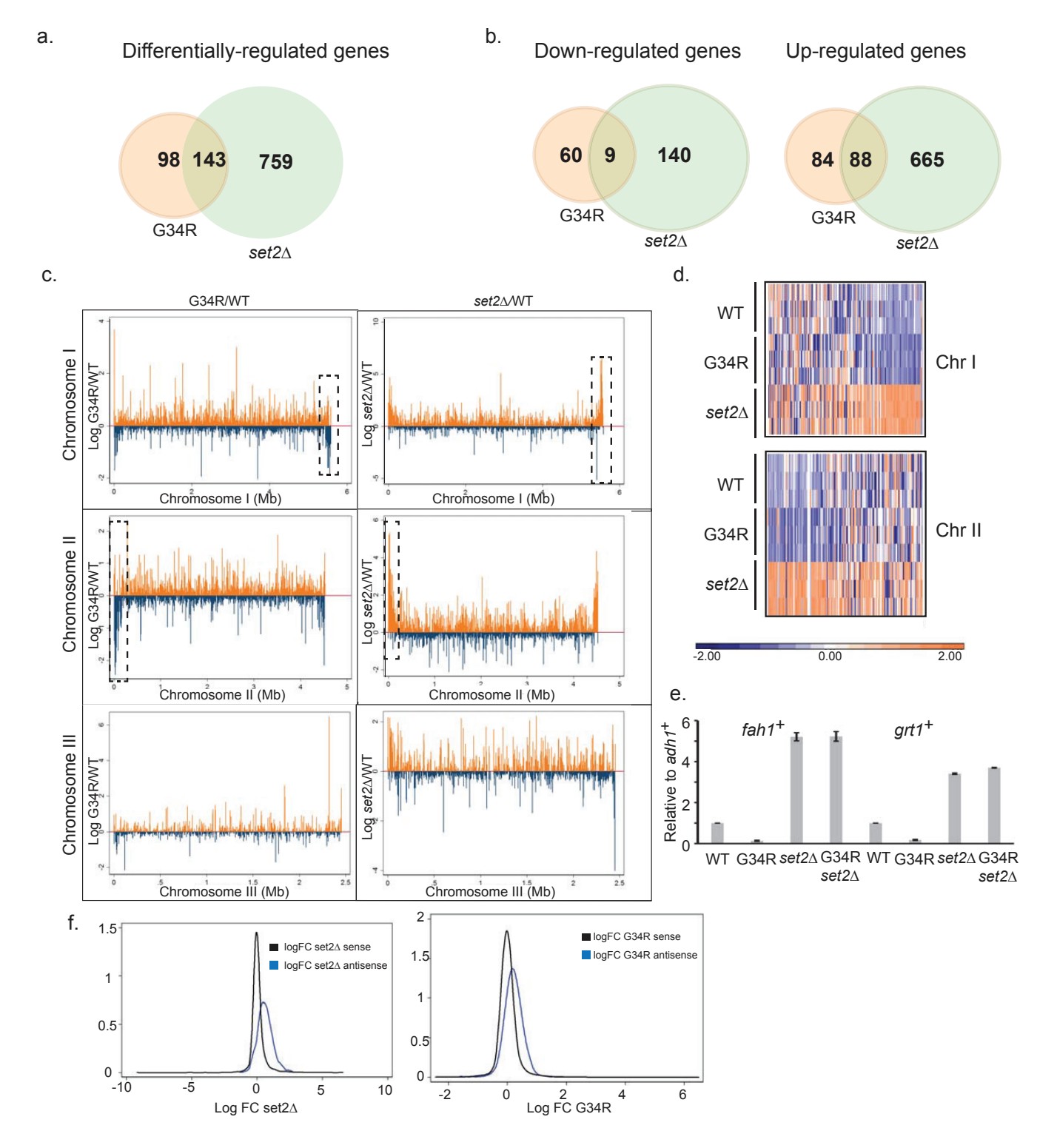

**Figure 2.** H3-G34R mutants show a distinct transcriptional profile to *set2Δ* cells. Schematic of genes that are (a) differentially-regulated, or (b) similarly down-regulated or up-regulated comparing either H3-G34R or *set2Δ* cells with H3-WT cells in RNA-seq analyses. Triplicate biological replicates were analyzed with cut-offs of 1.5 fold differences in expression, and false discovery rates of 5%. (c) RNA-seq profiles for chromosomes I, II, and III comparing Log fold change ratios for H3-G34R/H3-WT or *set2Δ*/H3 WT plotted against chromosome coordinates. (d) Zoomed-in regions of RHS Chr I (5.3 Mb-end, top) and LHS Chr II (first 300 Kb, bottom) as depicted in the boxed regions in (c) showing log FC data for individual biological replicates. (e) RT-PCR

*Figure 2 continued on next page*

*Figure 2 continued*

validation of *fah1+* and *grt1+* expression relative to *adh1+* expression. (**f**) Density plots of RNA seq reads for sense (black) and antisense transcripts (blue) against log fold change ratios for *set2Δ*/H3 WT (left) and H3-G34R/H3 WT (right). Please see *Figure 2—figure supplement 1* for analysis of heterochromatic loci, and *Supplementary file 3* for full RNA-seq analysis results.

The following figure supplement is available for figure 2:

**Figure supplement 1.** Heterochromatin integrity is maintained in H3-G34R cells.

*set2Δ* cells, upregulate antisense transcripts, indicative that trimethylation of K36 may repress antisense transcription.

In conclusion, interpretation of the transcriptional profiling results is complex. There is some overlap with Set2-mediated control as shown by similar regulation of some sense and some antisense genes, but also differences as evidenced by opposite regulation of transcription with sub-subtelomeric domains. We conclude that suppression of antisense gene transcription appears linked to trimethylation of K36, as it is reduced in both H3-G34R and *set2Δ*, and that repression of sub-subtelomeric domains may be linked to dimethylation of K36 in H3-G34R, as it is abolished in *set2Δ* or compound *set2Δ* H3-G34R mutants that lack dimethylation of K36.

## Heterochromatin is maintained in H3-G34R cells, but cells exhibit genomic instability

RNA-Seq provides information on unique transcripts. To determine whether constitutive heterochromatin that forms on repetitive sequences was affected by H3-G34R mutation, we monitored transcripts from pericentromeric loci and the subtelomeric *tlh* genes in H3-WT and H3-G34R cells (*Figure 2—figure supplement 1a*), with *clr4Δ* serving as a control for loss of heterochromatin (*Ekwall et al., 1996*; *Allshire et al., 1995*). Transcript levels were similar in H3-G34R and H3-WT, suggesting that heterochromatin is intact in H3-G34R mutants, whereas *set2Δ* showed a slight upregulation in subtelomeric *tlh* transcripts.

Additionally we monitored H3K9me2, a conserved heterochromatin mark, and the Swi6[HP1] protein which binds to that mark (*Nakayama et al., 2001*; *Bannister et al., 2001*). As shown in *Figure 2—figure supplement 1b and c*, H3-G34R mutation had no impact on H3K9 methylation or Swi6[HP1] recruitment to heterochromatic regions (*Ekwall et al., 1995*). ChIP against Cnp1 (*Castillo et al., 2007*), the fission yeast CENP-A protein that underlies and supports kinetochore assembly, showed no change in Cnp1 recruitment to centromeric central core sequences in H3-G34R mutant cells (*Figure 2—figure supplement 1d*). Together, the lack of apparent effect on kinetochore or heterochromatic structures indicates that the architecture of centromeric and telomeric heterochromatin is intact in H3-G34R cells.

We next tested genomic stability in H3-G34R mutants by monitoring the frequency of chromosome loss using a non-essential minichromosome Ch16 (*Niwa et al., 1989*). As expected, H3-WT cells showed low levels of loss of Ch16 (0.9% cells) (*Mellone et al., 2003*) and cells lacking the heterochromatin protein Swi6[HP1] displayed high frequencies of Ch16 loss (6.4%, *Figure 3a*). Interestingly, we found that H3-G34R cells lost minichromosomes at an elevated frequency (3.5%), while minichromosome loss in *set2Δ* was not significantly upregulated (1.6%). We further tested for specific chromosome segregation defects in H3-G34R strains by monitoring frequencies of lagging chromosomes on late anaphase spindles, using anti-tubulin antibodies to stain the spindle, and DAPI to stain DNA (*Figure 3b*) (*Ekwall et al., 1995*). Counting only late anaphase cells (spindle >10 microns), we found that chromosomes mis-segregated in 7.4% of H3-G34R cells compared to 0.9% in H3-WT, 1.3% in *set2Δ* and 25.6% in *clr4Δ* cells that lack heterochromatin. Furthermore, in cells synchronized using an *nda3-KM311* early mitotic block and release (*Hiraoka et al., 1984*), and stained for tubulin and DNA, ~38% of H3-G34R cells showed stretching of DNA or aberrantly positioned DNA on mid to late anaphase spindles compared with ~14% of H3-WT cells (*Figure 3c*). Together, these data show that the H3-G34R cells exhibit elevated chromosomal instability, which is independent of heterochromatin or kinetochore defects. The nature of the chromosome segregation defects in H3-G34R cells is suggestive of a defect in DNA condensation or resolution of sister chromatids after DNA replication (*Saka et al., 1994*).

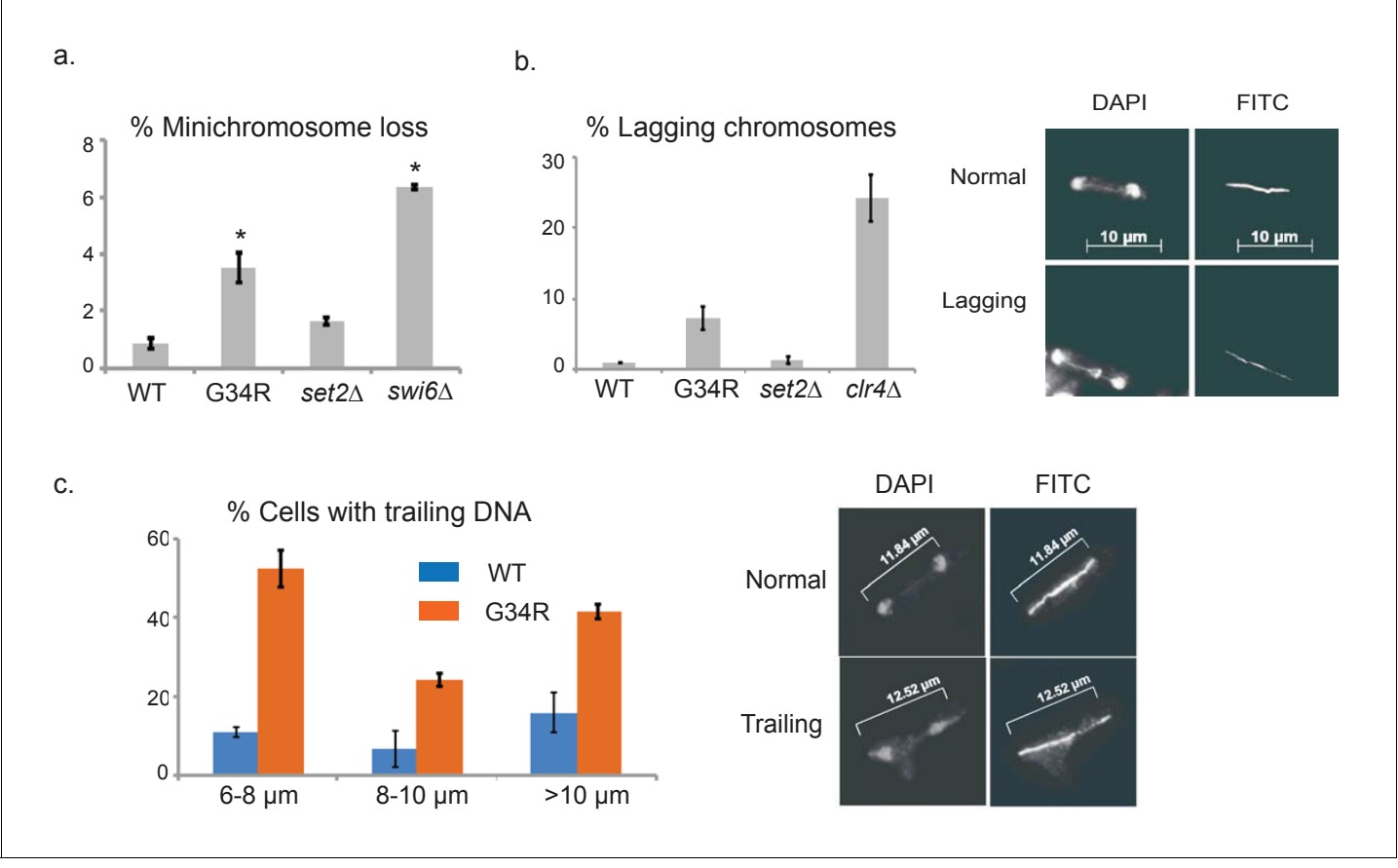

**Figure 3.** H3-G34R cells exhibit genomic instability. (a) Frequency of cells that lose the non-essential minichromosome Ch16 in H3-WT, H3-G34R, *set2Δ* and *swi6Δ* cells. Asterisk indicates significant difference from H3-WT (p<0.05). Mean ± SEM from 4 experiments shown. (b) Frequency of late anaphase cells that show a lagging chromosome in H3-WT, G34R, *set2Δ*, and *clr4Δ*. Mean ± SEM from 2 experiments shown. G34R and *clr4Δ* have small p values (0.1) although not significantly different from WT (c) Frequency of cells with chromosome segregation defects in H3-WT and H3-G34R cells. Cells were synchronized using *nda3-KM311* and chromosome segregation phenotypes were scored in cells with different spindle lengths. Data are represented as mean ± SD from 2 biological replicates. Right panels depict representative images of (b) normal and lagging or (c) 'trailing' chromosomes (DAPI = DNA; FITC = tubulin).

## H3-G34R mutants show DNA damage sensitivity upon replication stress

Since H3-G34R cells show increased chromosomal instability, and are reduced in histone H3 K36 acetylation and tri-methylation, which are modifications that are linked to DNA damage responses (*Pai et al., 2014*; *Pfister et al., 2014*; *Jha and Strahl, 2014*; *Li et al., 2013*), we tested H3-G34R strains for sensitivity to DNA damaging agents. Cells were plated on media containing hydroxyurea (HU; a ribonucleotide reductase inhibitor that depletes dNTPs), camptothecin (CPT; an agent that blocks topoisomerase 1 ligase activity), or methyl methanesulfonate (MMS; an alkylating agent) (*Figure 4a*). At the concentrations used, each of these agents predominantly affects DNA replication (HU and MMS), or requires DNA replication to inflict DNA damage (CPT). We found that chronic exposure to each genotoxin resulted in significant sensitivity of H3-G34R cells compared with H3-WT cells. Conversely, when H3-G34R and H3-WT cells were exposed to gamma irradiation (γIR) or bleomycin, two replication-independent genotoxins, H3-G34R mutants showed no sensitivity, whereas *set2Δ* cells were sensitive to γIR and bleomycin (*Figure 4b and c*) as seen previously (*Pai et al., 2014*). Thus H3-G34R, but not *set2Δ* cells appear selectively sensitive to agents that generate DNA replication stress.

To further address the relationship of the H3-G34R mutant with Set2, we tested genetic interactions between *set2Δ* and H3-G34R. In the absence of damage, there were no synthetic growth defects in the *set2Δ* H3-G34R double mutant. *set2Δ* were more sensitive than H3-G34R to MMS and

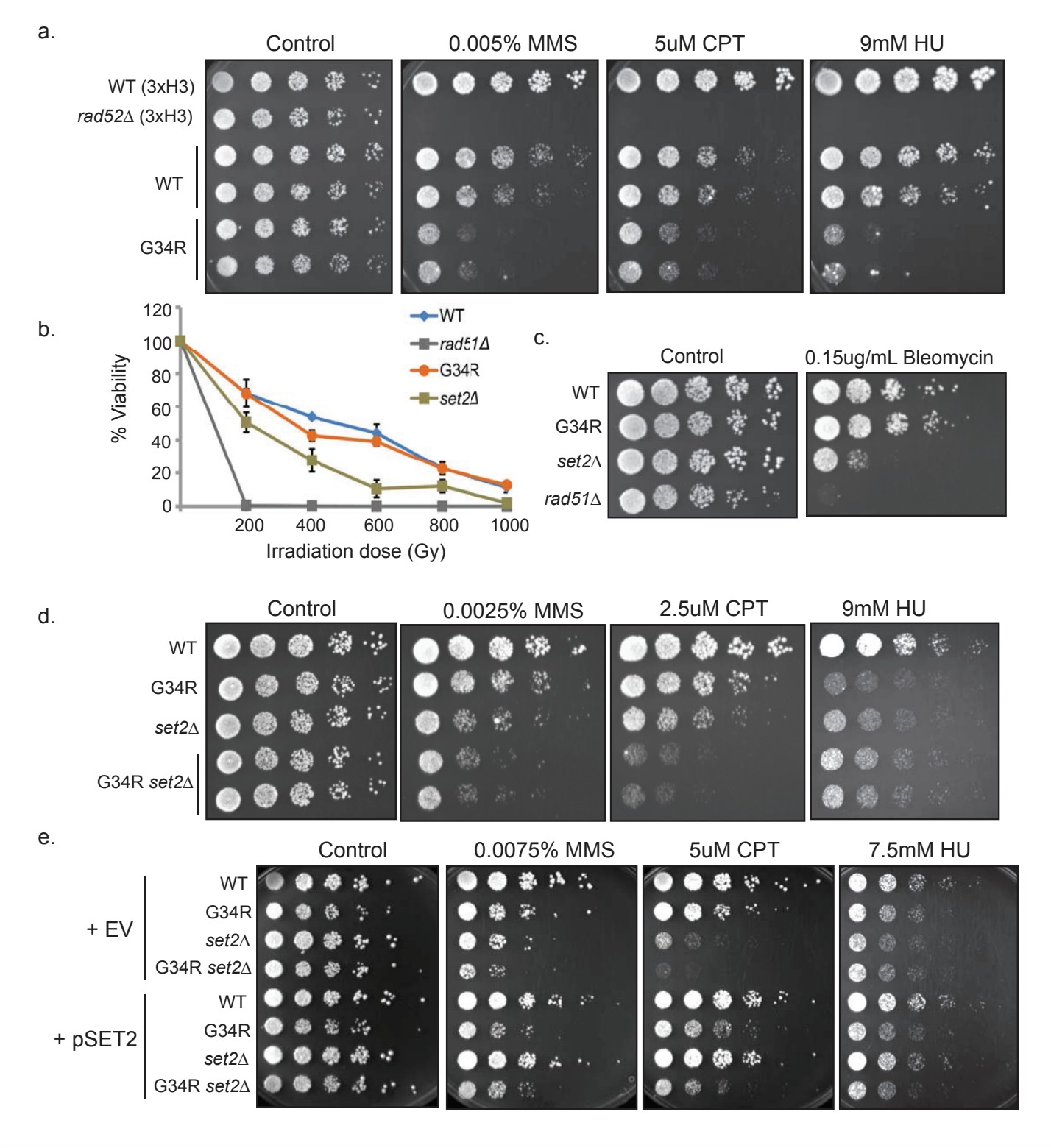

**Figure 4.** H3-G34R mutants show DNA damage sensitivity upon replication stress. (a) 5-fold serial dilutions showing the effect of methyl methanesulfonate (MMS), camptothecin (CPT), and hydroxyurea (HU) on the indicated strains. (b) Effect of γ−irradiation (IR) exposure on viability of H3-WT, H3-G34R, *set2Δ*, and *rad51Δ* cells. Data represent mean ± SEM from 2 biological replicate experiments. (c) Serial dilution assay showing the effect of bleomycin on the indicated strains. (d) Serial dilution assay showing the genotoxin sensitivity of H3-WT, H3-G34R, H3-WT *set2Δ*, and H3-G34R *set2Δ*

*Figure 4 continued on next page*

*Figure 4 continued*

cells upon MMS, CPT, or HU treatment. (**e**) Genotoxin sensitivity of indicated strains containing either an empty vector (EV) or a vector overexpressing Set2-3xFLAG (pSet2).

CPT, but showed similar sensitivity to H3-G34R on HU. Additive growth defects were seen for *set2Δ* H3-G34R double mutants on CPT and MMS, but not with HU (*Figure 4d*). Repair of damage from CPT or MMS-induced lesions likely involves the use of Holliday junctions, whereas recovery from HU-induced stress generally does not (*Branzei and Foiani, 2008*). The enhanced sensitivity of double mutants to CPT and MMS may indicate that *set2Δ* and H3-G34R mutants are defective in distinct aspects of HR-mediated repair requiring assembly of Holliday junctions. We further asked whether overexpression of Set2 could compensate for the DNA damage sensitivities of H3-G34R cells. Whereas pSet2-FLAG overexpression fully compensated for *set2Δ* growth defects on damaging agents, no growth advantage was conferred on H3-G34R cells (*Figure 4e*). However, since Set2 overexpression cannot compensate for H3K36me3 in H3-G34R cells (*Figure 1e*), it is unclear whether the DNA damage sensitivity of H3-G34R cells is linked to the lack of efficient tri-methylation on K36.

## H3-G34R cells have intact DNA replication checkpoints

We have shown that H3-G34R mutants are sensitive to chronic exposure to replication stress. DNA damage checkpoint signaling is conserved in fission yeast (*Figure 5a*) (*Melo and Toczyski, 2002*). Following DNA damage, the upstream sensor kinases Rad3 (ATR) and Tel1 (ATM) phosphorylate histone H2A on a C-terminal serine to generate γH2A and activate the downstream DNA damage checkpoint kinases Chk1 (*Capasso et al., 2002*) and the replication checkpoint kinase Cds1 (*Lindsay et al., 1998*).

To address whether DNA damage checkpoint signaling is functional in H3-G34R cells, we first monitored phosphorylation of H2A (γH2A). Western analyses of cells treated with HU or MMS showed no differences in γH2A phosphorylation levels in H3-G34R or *set2Δ* strains compared to H3-WT, indicative that the function of upstream kinases is not perturbed in H3-G34R cells (*Figure 5b*). Next, we monitored phosphorylation-related mobility shifts of Chk1-HA and found that Chk1 was efficiently activated by MMS treatment in H3-G34R cells, suggesting that DNA damage checkpoint signaling was normal (*Figure 5c*) (*Walworth and Bernards, 1996*). To assess Cds1 replication checkpoint function, we performed HU treatment of H3-G34R cells since, in *cds1Δ* cells, prolonged DNA synthesis occurs both during HU treatment and after release. This causes an accumulation of single stranded DNA (ssDNA) that can be visualized as large foci of replication protein A (RPA) (*Sabatinos et al., 2012*). Cells were arrested in HU, and RPA foci monitored 6 hr post-release. In contrast to the 'flares' of RPA seen in *cds1Δ* after release, H3-G34R showed no accumulation of RPA foci (*Figure 5d*), supporting that the Cds1 pathway is functional in H3-G34R mutants.

Finally, we tested H3-G34R in combination with checkpoint mutants for their ability to arrest after a prolonged exposure to genotoxin (HU), which can be monitored by elongation of the cells as an indication of proper checkpoint arrest. Control cells lacking both checkpoints (*cds1Δ chk1Δ*) failed to arrest after HU treatment, as indicated by their short length and entry into lethal mitoses with incompletely replicated chromosomes (*Figure 5e*) (*Lindsay et al., 1998*). Cells lacking either *cds1* or *chk1* elongated properly, likely arrested by compensation of the other kinase. H3-G34R *cds1Δ* and H3-G34R *chk1Δ* double mutants also arrested normally as elongated cells, suggesting that both checkpoints are intact in H3-G34R (*Figure 5e*). In summary, DNA damage checkpoint signaling appears to be functional in H3-G34R cells.

## H3-G34R cells show defects in homologous recombination pathways

Thus far, we have shown that H3-G34R mutants exhibit genomic instability that does not involve disruption of heterochromatin (*Figure 3*), that they are sensitive to DNA replication stress (*Figure 4*), and have functional DNA damage checkpoint signaling (*Figure 5*). We next assessed DNA damage repair pathways in H3-G34R cells (*Figure 6a*). First we asked how cells recover from replication stress. After HU treatment, completion of DNA replication occurs without the need for Origin Replication Complex (ORC) activity. However, some mutants that are sensitive to replication stress, such

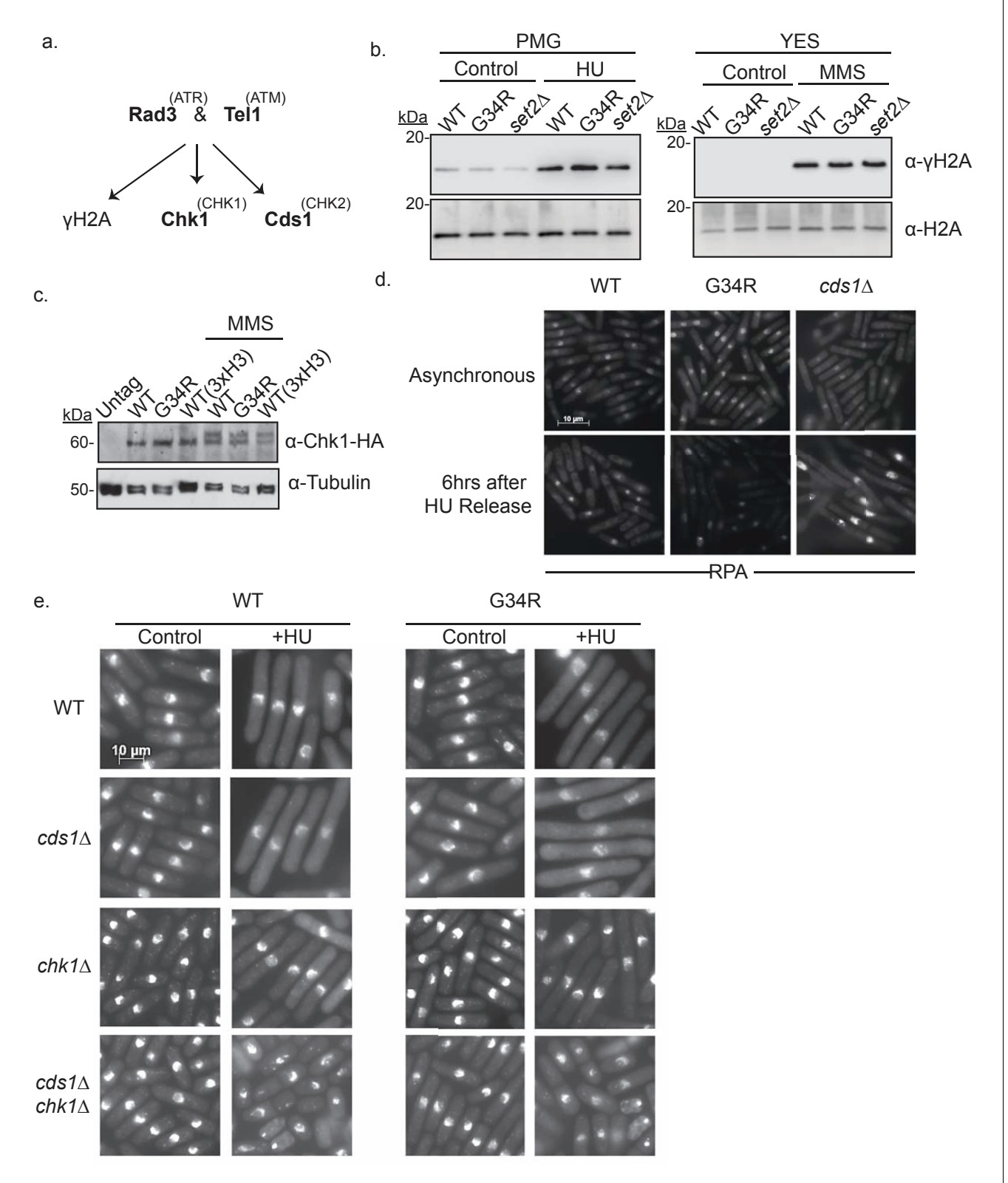

**Figure 5.** H3-G34R cells have intact DNA replication checkpoints. (a) Scheme depicting major kinases involved in checkpoint control and in brackets, their mammalian homologs. (b) Western blot analysis of γ-H2A phosphorylation in WT, G34R, and *set2Δ* cells. Cells were collected ±4 hr treatment with 15 mM HU or 0.05% MMS. Total H2A serves as a loading control. (c) Western blot showing Chk1-HA activation. Chk1-HA shows phosphorylation-dependent mobility shift in 0.05% MMS treated H3-WT and H3-G34R cells. Cells with all three H3/H4 genes (3xH3) serve as control. Tubulin is used as a

*Figure 5 continued on next page*

*Figure 5 continued*
loading control. (**d**) RPA (Rad11-GFP) immunofluorescence of indicated strains in either untreated conditions or after 6 hr release following 4 hr of 11 mM HU treatment. (**e**) Immunofluorescence imaging of DAPI stained cells for analysis of checkpoint-dependent cell elongation after treatment with 11 mM HU for 4 hr. Combinations of checkpoint mutants (*cds1Δ*, *chk1Δ*, or *cds1Δ chk1Δ*) with either H3-WT or H3-G34R mutations were assessed.

as a mutant of the PTIP homolog (*Cho et al., 2003*), *brc1Δ*, require ORC to recover from HU arrest (*Bass et al., 2012*). We tested whether H3-G34R cells depend on ORC to recover from HU, by use of a temperature sensitive allele in ORC, *orp1-4*. Cells were synchronized with HU at 25°C, and released at either 25° or 36°C for 4 hr prior to plating (*Figure 6b*). As expected, any *orp1-4* cells that are released to 36°C die (green and purple bars), whereas *orp1-4* H3-WT cells that are HU arrested and released at 25°C survive. Intriguingly, unlike *orp1-4 brc1Δ* which die due to severe mitotic defects (*Bass et al., 2012*) (red bars, *Figure 6b*), we found that *orp1-4* H3-G34R survive. Thus, H3-G34R cells do not rely on ORC to recover from HU-imposed replication stress.

Next, we systematically mutated genes that are key to well defined DNA repair pathways in H3-G34R and H3-WT cells. We mutated genes for post-replication repair (PRR), non-homologous end joining (NHEJ), and homologous recombination (HR) in H3-WT and H3-G34R cells (*Figure 6c–e*). To test PRR, we generated combination mutants of H3-G34R with either *rad18* (important for trans-lesion synthesis) or *ubc13* (important for template switching) (*Branzei et al., 2008*). H3-G34R mutants with either of these mutations show synergistic sensitivity to MMS, suggesting that the PRR pathways are functional in H3-G34R cells (*Figure 6c*). Additionally we monitored epistasis with a mutant in the NHEJ pathway. Combination mutants of H3-G34R were generated with mutants in Ku70, a subunit of the Ku complex responsible for recognizing double strand breaks (DSBs) (*Manolis et al., 2001*). H3-G34R showed synergistic defects with *ku70Δ* when assessed on MMS compared with either single mutant, indicating that NHEJ is functional in H3-G34R cells (*Figure 6d*).

Furthermore, we assessed the genotoxin sensitivity of H3-G34R in combination with a mutant necessary for HR-directed repair. Combination mutants were generated with *rad51Δ* (Rad51 encodes a strand exchange protein that forms a filament on DNA and facilitates the search for homology). In contrast to mutations with PRR or NHEJ proteins, H3-G34R *rad51Δ* double mutants showed epistasis with single mutants when plated on MMS (*Figure 6e*). This suggests that H3-G34R cells have a defective HR pathway.

To extend this analysis, we developed an assay to directly monitor the frequency of HR-directed repair (*Figure 6—figure supplement 1a*). We transformed *leu1-32* mutant cells (that bear a single nucleotide mutation in *leu1* that renders cells auxotrophic for leucine) with a 576 bp fragment of wild type *leu1+* and scored *leu1+* transformants that can only arise by HR, normalizing for transformation efficiencies of strains using plasmid DNA (*Figure 6f*). As expected, *rad51Δ* cells were very defective with HR rates of only ~3% of H3-WT cells. H3-G34R and *set2Δ* strains were also defective (~35% efficiency) and the double H3-G34R *set2Δ* mutant showed an even stronger defect, with a 90% drop in HR efficiency. The reduction in HR in *set2Δ* was not due to the genetic background, as we obtained similar results when *set2Δ* was assessed in a 3xH3/H4 genetic background (*Figure 6—figure supplement 1b*). Thus both H3-G34R and *set2Δ* reduce HR efficiency but via distinct mechanisms, as additive defects are seen in the *set2Δ* H3-G34R double mutant. We note that our finding of an HR defect in *set2Δ* differs from the previously published result of enhanced HR and defective NHEJ in 3xH3/H4 *set2Δ* cells as described in *Pai et al., 2014*. Their study utilized an engineered non-essential minichromosome with an HO break site. The authors monitored four different markers for loss, and interpret combinations of marker loss as NHEJ-repair, gene conversion, loss of heterozygosity and loss of the chromosome. However, due to the nature of the assay, there are many possible causes of apparent marker loss which may cloud interpretation. Our simple and direct approach to measure HR relies on a site specific gain of marker function at a defined locus. We note that we have just assessed HR at one site, and it is quite feasible that distinct outcomes may be obtained by assessing HR at different genomic regions. We sought to determine if the difference in interpretation of Set2 function was indeed caused by the different assays or by strain differences between the *set2Δ* backgrounds. We performed the transformation-based HR assay in the *set2Δ* strains used in *Pai et al., 2014*, and in NHEJ defective *ku70Δ* cells (*Figure 6—figure supplement 1c*), and found that all *set2Δ* strains tested showed a defect in HR, whereas HR efficiency was

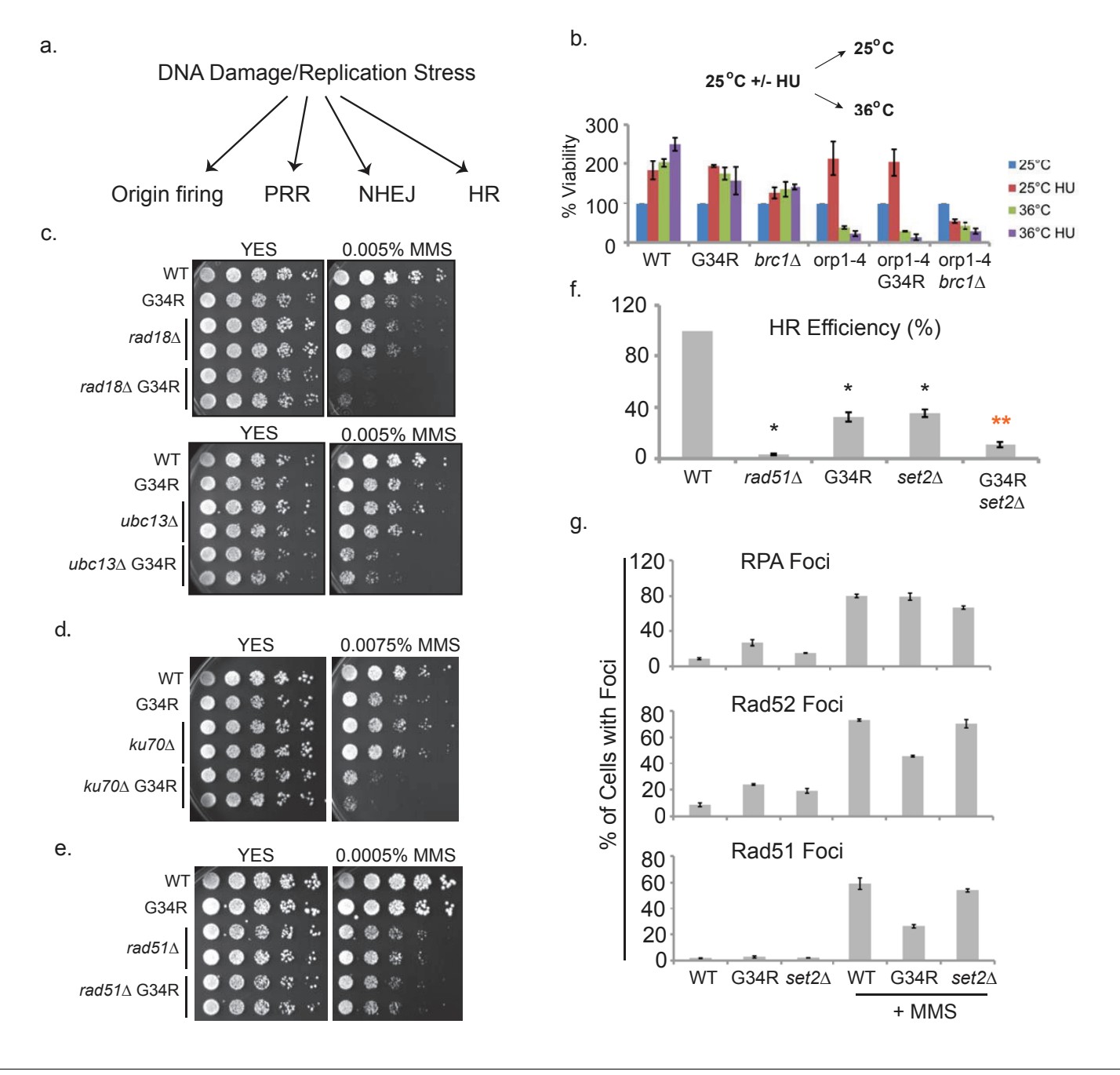

**Figure 6.** H3-G34R cells show defects in homologous recombination pathways. (a) Scheme depicting DNA damage repair pathways. (b) Cells of indicated genotypes were cultured at 25°C in the presence or absence of 11 mM HU for 6.5 hr and then washed and either kept at 25°C (permissive) or shifted to 36°C (restrictive) for 4 hr. Cells were then plated at 25°C to determine cell viability. Numbers were normalized to untreated 25°C control values for each genotype, and results were averaged from 2 independent biological replicates. (c–e) Serial 5-fold dilution assays of H3-WT or H3-G34R cells and double mutants with (c) PRR pathway mutants *rad18Δ* or *ubc13Δ*, (d) NHEJ mutant *ku70Δ* or (e) HR mutant *rad51Δ* plated on media ± MMS treatment. (f) Homologous recombination assay based on correction of *leu1-32* mutation by HR. Cells of indicated genotypes were transformed with a *leu1* gene fragment to measure HR, or plasmid to measure transformation efficiency. Relative HR efficiency is shown as 100% for H3-WT, and results are averaged from 6 independent experiments with error bars representing SEM. Black asterisks reflect significant differences with H3-WT cells (p<0.05), and red asterisks, significant differences from both H3-G34R and *set2Δ* (p<0.01). See also *Figure 6—figure supplement 1*. (g) Percentage of cells that form foci of either RPA (Rad11-GFP), Rad52-GFP, or Rad51 before or after treatment with 0.05% MMS for 4 hr. Immunostaining with GFP antibodies was used for monitoring RPA and Rad52 while anti-Rad51 antibodies were used to detect Rad51. Cells were counted from 2 independent experiments with errors representing SEM. See also images in *Figure 6—figure supplements 1* and *2*.

*Figure 6 continued on next page*

*Figure 6 continued*

The following figure supplements are available for figure 6:

**Figure supplement 1.** Characterization of H3-G34R and *set2Δ* cells in homologous recombination-directed repair.

**Figure supplement 2.** FACS analysis of *cdc10-M17* arrested and released cells (**G1**) on the left, or mitotically arrested and released cells (*nda3-km311*) on right.

**Figure supplement 3.** Localization of homologous recombination-directed repair proteins in H3-G34R cells.

strongly enhanced in *ku70Δ*, as HR is now the sole method of DNA break repair available. We conclude that the difference in interpretation of whether *set2Δ* cells are deficient in HR lies with the use of distinct assays.

In fission yeast, NHEJ is restricted to G1, while HR operates in S and G2 phases of the cell cycle (*Ferreira and Cooper, 2004*). One possible cause of the decreased HR in H3-G34R cells and the dependence of H3-G34R cells on NHEJ for viability (*Figure 6d*) is an alteration in the cell cycle, with an extension of G1. We measured the generation time during exponential growth and found that H3-G34R had an extended doubling time of 201 min (±4 min) compared to 180 min (±10 min) for H3-WT, and 194 min (±8 min) for *set2Δ* in rich media at 32°C. To determine if the growth delay of H3-G34R mutants was linked to a particular cell cycle stage, we monitored cell cycle kinetics using FACS sorting of cells synchronized in G1 or in metaphase and then released back into the cycle at 25°C. As shown in *Figure 6—figure supplement 2*, *cdc10-M17* (G1) arrested and released cells begin DNA synthesis at 40 min after release. H3-WT cells have completed DNA synthesis by 80–100 min, whereas S phase is prolonged in H3-G34R cells and not completed until 140–160 min after release. It is difficult to draw conclusions from the *nda3-km311* arrest and release experiment since fission yeast do not undergo cytokinesis until G1, which complicates analysis of G1 cells. From these data we can surmise that H3-G34R cells spend longer in S phase when released from a G1/S block.

Next we asked whether there were defects in localization of HR proteins. We evaluated foci formation of RPA, Rad51, and Rad52 (*Figure 6g* and *Figure 6—figure supplement 3a*). Consistent with enhanced replication stress in H3-G34R cells, H3-G34R cells showed increased numbers of RPA and Rad52 foci (*Bass et al., 2012*). As expected, all cells showed increased foci for all three proteins following MMS treatment. Surprisingly however, fewer MMS-treated H3-G34R cells showed Rad51 and Rad52 foci compared with H3-WT and *set2Δ*. The reduction in foci was not linked to a change in protein expression as Rad51 and Rad52 protein levels were similar in MMS treated H3-WT and H3-G34R cells (*Figure 6—figure supplement 3b*). Additionally, Rad52 foci co-localized with RPA, suggesting that Rad52 foci occur at sites of DNA damage (*Figure 6—figure supplement 3c*). In summary, H3-G34R cells exhibit elevated marks of replicative stress, and on MMS-damage, have diminished foci formation for Rad51 and Rad52, which may be linked to the HR defect seen in H3-G34R cells.

We questioned whether the chromosome segregation defects in H3-G34R may be related to delayed resolution of replicative intermediates. To address this, we monitored RPA localization in anaphase cells following *cdc10* block and release, and found that H3-G34R cells exhibit anaphase bridges, with RPA marking ultrafine bridges between DAPI staining masses (*Figure 6—figure supplement 3d*). We are unable to determine the frequency of these bridges in anaphase cells since attempts to co-stain the spindle hampered visualization of bridges, but they were easy to identify in synchronized populations suggesting that they are frequent events. This supports that chromosome segregation defects in H3-G34R cells are linked to the delayed resolution of replication intermediates.

## H3-G34R mutants exhibit delayed recovery from replicative stress

HR-directed repair is often utilized to restart replication forks following stress. Because we observed a defect in HR, and a defect in Rad51 and Rad52 foci formation in MMS-treated H3-G34R cells, we speculated whether this was due to a delay in Rad51/Rad52 filament formation, and thus a defect in

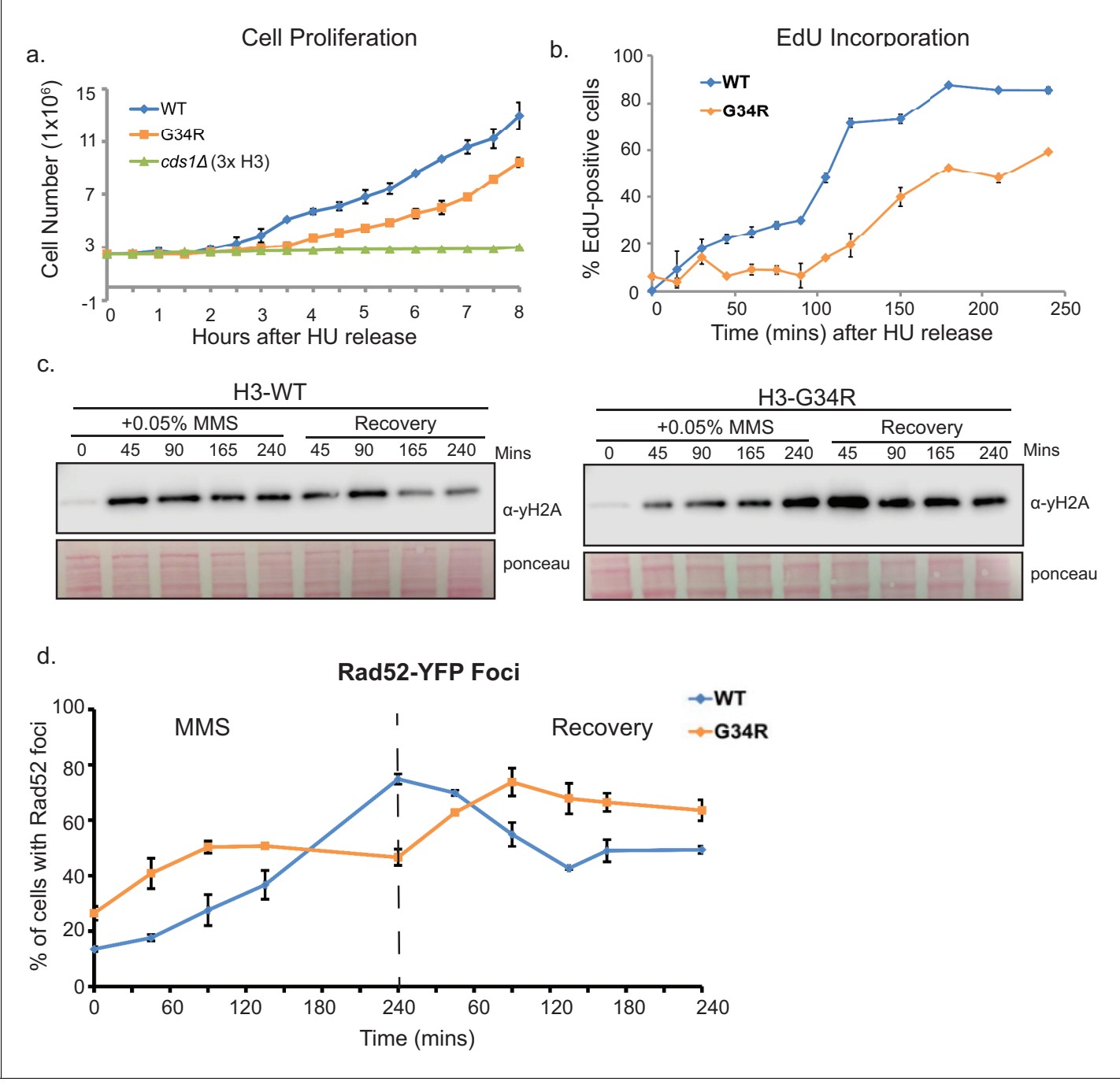

**Figure 7.** H3-G34R mutants exhibit delayed recovery from replicative stress. (a) Cell proliferation of H3-WT, H3-G34R, and *cds1Δ* cells was analyzed after synchronization in 11 mM HU for 4 hr. Cells were counted at 30 min intervals following release. Results represent average of 2 independent experiments, with error bars reflecting SEM (H3-WT and H3-G34R) and a single *cds1Δ* experiment. (b) Measure of EdU labeling of newly synthesized DNA. Cells were modified to incorporate nucleotide analogs. WT and G34R cells were synchronized in 11 mM HU for 4 hr and released into fresh media containing EdU and collected and ethanol fixed over a time course. EdU was labeled by ClickIT with Alexa Fluor 488 and samples were analyzed by FACS, gating on cells that fully incorporated EdU. Assay was performed twice, and each sample was read three times. Data from one experiment is shown, with values representing means of triplicate reads with background from control samples subtracted. Error bars represent SD. (c–d) Time course to analyze (c) γH2A by western blotting or (d) Rad52 (Rad52-GFP) foci formation in H3-WT and H3-G34R cells during treatment with 0.05% MMS and recovery. For (d), error bars represent SEM for 2 independent experiments, and the time course for γH2A westerns was repeated twice.

replication restart. To address whether the kinetics of recovery were affected, cells were arrested with HU, washed, and then counted at intervals following release (*Figure 7a*). Checkpoint-defective *cds1Δ* cells do not recover from the arrest. While H3-G34R do delay proliferation for longer than H3-WT (3 hr rather than 2 hr), the proliferation rates at later time points appear similar. To directly monitor the rate of DNA replication, we arrested cells with HU and on release, monitored de novo DNA synthesis by EdU incorporation, gating on cells that had fully incorporated EdU (*Figure 7b*). Interestingly, de novo DNA synthesis is initially delayed in H3-G34R cells but at later time points after HU release, the rates of DNA synthesis appear similar in H3-WT and H3-G34R cells.

To further evaluate the kinetics of DNA damage response, we monitored γH2A phosphorylation (*Figure 7c*) and counted cells that bear Rad52 foci (*Figure 7d*) during MMS treatment and recovery. MMS treatment led to a rapid induction of γH2A signal in H3-WT cells, such that at 45 min treatment, maximal signal was achieved which was retained during treatment. Surprisingly, γH2A signal accumulated more slowly in H3-G34R cells, and peaked at 45 min into the recovery period, at higher levels than seen in H3-WT cells. On recovery from MMS, γH2A remained intensely phosphorylated at later time points in H3-G34R cells compared with H3-WT cells (see *Figure 7c*).

Similar to γH2A, Rad52 foci accumulated faster in MMS treated H3-WT cells than H3-G34R, even though more Rad52 foci were present in untreated H3-G34R cells than in H3-WT (*Figure 7d*). During recovery, H3-WT cells with Rad52 foci dropped to ~50% by 180 mins while Rad52 foci continued to accumulate in H3-G34R, peaking at 90 min following release, before dropping. These data suggest that the kinetics of DNA damage response are different between H3-WT and H3-G34R cells, with H3-G34R mutants showing a slower induction of damage response, as well as accumulation of recombination proteins at later timepoints following damage, which may contribute to the HR-mediated repair defect in H3-G34R mutants.

## Discussion

In this study, we generated single copy histone mutants in fission yeast with the aim of studying the behavior of the H3-G34R mutant. This system allows a detailed examination of the possible consequences of this histone mutant, divorced from the complexity of mammalian histone regulation. Using this system, we show that H3-G34R mutants have focal histone modification changes, with reduced tri-methylation and acetylation, but retention of di-methylation on the nearby K36 residue on histone H3. These results are consistent with some perturbation of Set2 function and we demonstrate that this defect cannot be overridden by overexpression of Set2 as, even under these conditions, H3-G34R cells fail to accumulate normal levels of H3K36me3.

Furthermore, the transcriptional consequence of H3-G34R mutation is limited, with fewer genes affected than in *set2Δ*. Indeed for genes whose transcription is altered in both *set2Δ* and H3-G34R, only some are coordinately regulated. Notably, these include antisense genes, which suggests that trimethylation of H3K36 is important for suppression of antisense gene transcription. Interestingly, genes showing discordant regulation map to specific chromosomal loci- the ST regions. Transcriptional upregulation in ST domains in *set2Δ* is likely a direct consequence of loss of Set2 function on H3K36, since the histone H3 K36A mutant shows similar upregulation of expression of ST domains (*Matsuda et al., 2015*). Of interest, ST regions have recently been shown to be the most highly condensed regions of the genome, forming 'knobs', with condensation linked to Set2 function (*Matsuda et al., 2015*), but the biological implications of knob formation are not yet known.

Perhaps the most consequential findings from this work are the effects of H3-G34R mutation on genome stability. We see evidence for enhanced chromosome loss and for defects in chromosome segregation, with accumulation of cells that have trailing or 'stretched' chromosomes. We find that RPA coats ultrafine anaphase bridges in mitotic H3-G34R cells, marking DNA that links chromosome masses, which is indicative of unresolved replication problems. These data are consistent with our finding that H3-G34R cells are sensitive to replication stress, and that HR pathways are defective in these cells. These phenotypes are unlikely to be due to transcriptional deregulation in H3-G34R cells since transcription of DNA damage genes, under non-perturbed conditions, is unaltered. Instead, as summarized in *Figure 8*, these findings may derive from the altered kinetics of signal response to damage in H3-G34R cells, with slowed accumulation of γH2A and delayed formation of Rad52 foci, features which may also contribute to the delayed replication of H3-G34R cells during recovery from

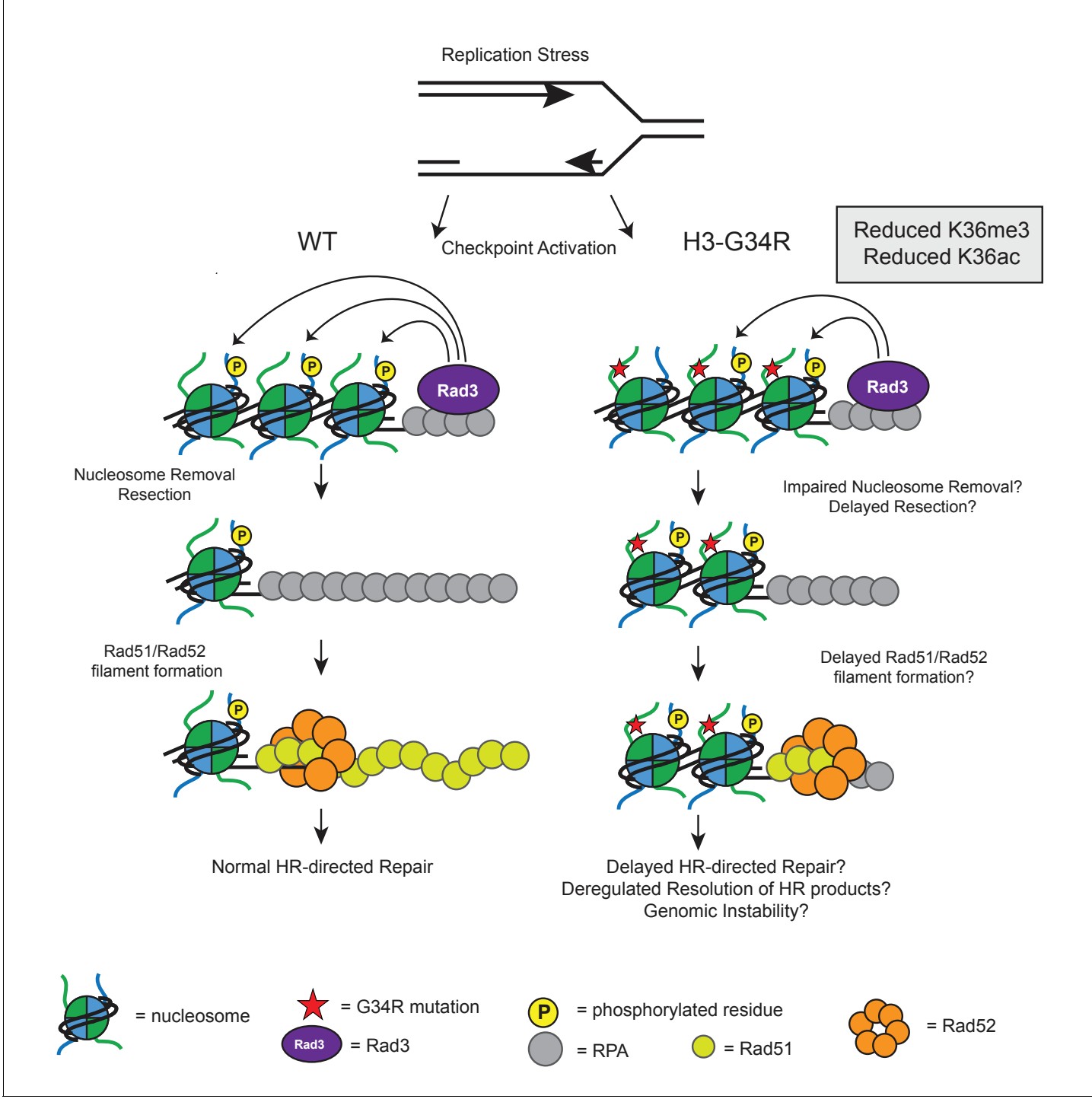

**Figure 8.** Model illustrating how HR defects in H3-G34R mutants may compromise genomic integrity. Model illustrates the key defects in H3-G34R cells and how these may contribute to genomic instability. Replication forks provide an endogenous source of DNA damage, as they are fragile structures that are prone to stalling and collapsing. (*left*) To cope with replication stress, cells activate the checkpoint signaling pathway by sensing ssDNA via Rad3. Rad3 interacts with ssDNA through RPA and phosphorylates H2A, which helps to transduce the checkpoint response to downstream kinases. For lesions that require HR-directed repair, nucleosome removal and subsequent 5' to 3' DNA resection exposes ssDNA which is initially coated with RPA, and then replaced by Rad51, facilitated by the Rad52 protein. The search for a homologous template can begin once sufficient Rad51-ssDNA filament has been formed and strand invasion of the Rad51-ssDNA filament generates Holiday junctions which need to be resolved prior to cell division. (*right*) In H3-G34R cells, HR is defective. Checkpoint signaling is functional, but appears delayed with slower accumulation of γH2A possibly due to impaired nucleosome removal and delayed resection, which would limit the amount of Rad3 on RPA/DNA. Retention of γH2A indicates a defect in nucleosome

*Figure 8 continued on next page*

*Figure 8 continued*

removal and resection which in turn, could account for the delayed Rad52 foci formation in H3-G34R cells and likely delayed Holiday junction resolution following repair. H3-G34R cells show elevated chromosome segregation defects without defects in heterochromatin or kinetochore chromatin, suggestive that a delayed resolution of HR products contributes to genomic instability.

replication stress. H3-G34R mutation may thus impair HR-mediated repair of DNA damage induced DSBs.

How might H3-G34R mutation elicit these effects? One possibility may be via modulation of H3K36 methylation signal. However this seems unlikely given that cells that lack Set2 show no defects in chromosome segregation, and the HR defects that we uncovered in *set2Δ* cells are not epistatic to HR defects in H3-G34R cells. Another alternative is that the altered kinetics for γH2A phosphorylation and Rad52 foci formation during MMS treatment and recovery are indicative of chromatin defects in the H3-G34R cells. H2A phosphorylation occurs within the C-terminal tail of H2A, which projects out from the DNA and occupies a territory close to the N-terminal tail of H3 within the nucleosome structure (*Du and Briggs, 2010*). Incorporation of an additional basic charge in the H3 tail close to where the H3 tail exits the H3 core domain may have conformational consequences and the H3-G34R tail may modulate kinase activity on the C-terminal H2A tail (*Figure 8*). However, checkpoint signaling is clearly functional in H3-G34R cells, even if it is delayed (*Figure 5*). Once damage has occurred, H3-G34R cells retain γH2A for longer than H3-WT (*Figure 7c*). It is thought that, in order for γH2A to be dephosphorylated, nucleosomes must first be removed from chromatin where they can be acted upon by phosphatases in free nuclear pools (*Keogh et al., 2006*; *Jablonowski et al., 2015*). Retention of γH2A may stem from a defect in efficient chromatin remodeling, thereby preventing timely repair. In support of this, an H2A mutant that mimics γH2A shows decreased interaction with the chromatin remodeler Fun30 (*Eapen et al., 2012*).

H3-G34R cells show defects in HR-directed repair. Nucleosome removal from damaged chromatin facilitates resection and subsequent coating by RPA. After sufficient resection of DNA, Rad52 helps to facilitate the exchange from RPA-coated DNA to a Rad51-coated filament, to allow for homology searching and HR repair. The presence of a γH2A mimetic can inhibit the rate of resection (*Eapen et al., 2012*). Here, we show that H3-G34R mutants are specifically deficient for HR (*Figure 6e*) with prolonged retention of γH2A and delayed Rad52 foci formation (*Figure 7c,d*). It is possible that H3-G34R cells are defective in timely nucleosome removal and resection (see *Figure 8*). Following this model, delayed HR-mediated repair in H3-G34R cells, or delayed resolution of replicative intermediates in undamaged H3-G34R cells, may allow accumulation of unresolved Holliday junctions prior to mitosis, and contribute to the chromosome segregation defects shown in this study (*Figure 3* and *Figure 6—figure supplement 3d*). We note that consistent with our findings, clearance of γH2A.X foci in human cells that lack H3K36me3 due to deletion of SETD2 is delayed (*Hacker et al., 2016*), and such cells are defective for HR and exhibit genomic instability (*Pfister et al., 2014*).

Clearly an important consideration is that our data was obtained in an organism where the whole population of histone H3 is mutant. In pHGG, only one allele of four that encodes H3.3 protein is affected in cells with 13 other genes that encode wild type H3.1 and H3.2 proteins. Unlike the H3K27M mutation that dominantly blocks PRC2 activity, G34R mutation does not dominantly affect K36 methylation (*Lewis et al., 2013*; *Zhang et al., 2017*). Instead, we suggest that the H3.3 G34R mutant may be greatly enriched at specific chromosomal loci. H3.3 is deposited into regions of constitutive heterochromatin including repetitive telomeric and pericentromeric domains, and at regions of high transcriptional activity through the activity of specialized chaperones (reviewed in *Kallappagoudar et al., 2015*). These sites of H3.3 deposition are already subject to replicative stress. As cells traverse S phase, the replicative delay incurred by the presence of G34R mutant H3.3 at these sites and impaired resolution of replication intermediates could greatly enhance genomic instability and offer a potent source of genomic perturbations to facilitate tumor development (*Macheret and Halazonetis, 2015*).

# Materials and methods

## Strain generation

Histone mutant strains were generated as described (*Mellone et al., 2003*). Gene mutagenesis used standard PCR-based procedures, and strains are listed in *Supplementary file 4*. All crosses used random spore analysis with nutritional/ drug selection and PCR verification (including verification of loss of additional H3/H4 genes) and sequencing of H3.2 allele. Two independent clones for each genotype were used in nearly all experiments, and all experiments were performed at least twice.

## Yeast growth media

Fission yeast were maintained on rich (YES), or *pombe* minimal with glutamate (PMG) media with appropriate supplements (*Moreno et al., 1991*). PMG is Edinburgh minimal medium with glutamate.

## Plasmid DNA

pSet2-3xFLAG vector for overexpression of Set2 (JP 2595) was generated by amplification of an endogenously tagged Set2 strain (PY 9101) using primers JPO-4159 and JPO-4160, and cloned into pREP41 (*Basi et al., 1993*) using Clontech in-fusion cloning kit.

JPO-4159: TTTGTTAAATCATATGTCGACATGCAGACGGCATCATCTCT,
JPO-4160: ACCCGGGGATCCTCTAGAACCTACAGGAAAGAGTTACTCAAGAATAAGAAT

## Histone purification for mass spectrometric studies (*Figure 1f*)

Histones were purified following a previously described purification protocol with some modification (*Sinha et al., 2010*). A 150 mL culture was inoculated to a density of $1.4 \times 10^6$ cells/mL in 4X YES media from a starter culture grown overnight in 4X YES at 25°C. The cells were grown to a density of $3.6 \times 10^7$ cells/mL and harvested by spinning at 3000 RPM for five minutes in a benchtop centrifuge. Cells were washed with $H_2O$ containing 10 mM sodium butyrate. The cells were then washed with NIB buffer (250 mM sucrose, 20 mM HEPES pH 7.5, 60 mM KCl, 15 mM NaCl, 5 mM MgCl2, 1 mM CaCl2, 0.8% Triton X 100, 0.5 mM spermine, 2.5 mM spermidine, 10 mM sodium butyrate, 1 mM PMSF, and Sigma yeast protease inhibitor). The pellet was frozen on dry ice and stored at −80°C. For lysis, the pellet was resuspended in 2 mL of NIB and transferred to a bead beater tube along with chilled acid washed glass beads. The sample and beads were frozen on dry ice and bead beaten for a total of 10 min at max power. The sample was collected by 'piggy backing' into a 50 mL Oak Ridge tube at 3000 RPM at 4°C in a benchtop centrifuge. The samples were then pelleted by centrifuging at 20,000 RPM for 10 min at 4°C in a Beckman Avanti centrifuge J-30I centrifuge using a JA25.50 rotor. The supernatant was discarded and the pellet was washed in 15 mL of NIB. The pellet was then resuspended in 10 mL of 0.4N $H_2SO_4$ and sonicated for one minute at max power before incubation for two hours on a rotating wheel at 4°C. The sample was pelleted by centrifuging at 20,000 RPM at 4°C for 10 min and the supernatant was transferred to a new tube along with 5 mL of 5% buffer G (5% guanidine HCl and 100 mM potassium phosphate buffer pH 6.8) where the pH was adjusted to 6.8 using 5N KOH. 0.5 mL of Bio-Rex pre-equilibrated in 5% buffer G was added to the sample and incubated at room temperature with rotation overnight. The resin was then washed 2X with 20 mL of 5% buffer G and incubated with 3 mL of 40% buffer G (40% guanidine HCl and 100 mM potassium phosphate buffer pH 6.8) for one hour at room temperature to elute the bound protein. Buffer exchange and concentration was performed against 5% acetonitrile with 0.1% TFA to a final volume of 150 µl and the sample was stored at −80°C. Protein concentration was measured and 10 µg of sample was run on a gel for quality control.

## UPLC−MS/MS analysis

Histone samples were TCA precipitated, acetone washed, and prepared for mass spectrometry analysis as previously described (*Kuo and Andrews, 2013*). A Waters (Milford, MA) Acquity H-class UPLC system coupled to a Thermo (Waltham, MA) TSQ Quantum Access triple-quadruple (QqQ) mass spectrometer was used to quantify modified histones. Selected reaction monitoring was used to monitor the elution of the acetylated and propionylated tryptic peptides. Transitions were created to study acetylation to *pombe* H3 wild-type and mutants as well as the H4 tails. The detailed

transitions for peptides of H3 that vary in sequence from *xenopus* are reported in *Supplementary file 2* and the peptides used are listed in *Supplementary file 1*; the transitions for the *xenopus* peptides have been previously reported (*Kuo et al., 2014*).

## QqQ MS data analysis

Each acetylated and/or propionylated peak was identified by retention time and specific transitions. The resulting peak integration was conducted using Xcalibur software (version 2.1, Thermo). The fraction of a specific peptide (Fp) is calculated as

$F_P = I_s / (\sum I_p)$, where $I_s$ is the intensity of a specific peptide state and $I_p$ is the intensity of any state of that peptide.

## Chromosome stability assays

*1. PFGE.* (*Figure 1—figure supplement 1–1e*) Cells were washed in ice-cold Stop Buffer (150 mM NaCl, 50 mM NaF,10 mM EDTA,1 mM NaN3) prior to processing for PFGE as described previously (*Andrews et al., 2005*). Samples were run on 0.8% chromosome-grade agarose (Bio-Rad) Tris-acetate-EDTA gels for 72 hr with a pulse time of 1800 s at 2 V/cm and an angle of 100°.

*2. Minichromosome loss.* (*Figure 3a*). The minichromosome (Ch16) (*Niwa et al., 1989*) bears an *ade6-216* allele which can complement the *ade6-210* allele present within the strain background. Loss of Ch16 causes loss of complementation of function of *ade6+* and accumulation of red pigment when cells are grown on limiting adenine media. Strains containing Ch16 were grown in PMG –Leu (to maintain Ch16) at 32°C to a density of $5 \times 10^6$ cells/mL. Cells were diluted in PMG (no additives) to a final concentration of $5-10 \times 10^3$ cells/mL. Cells were plated onto PMG agar supplemented with amino acids and nucleobases with limiting (10% normal concentration) adenine, incubated at 25°C for 5 days and then transferred to 4°C to let red color develop. Ch16 loss frequencies were calculated by counting half sectored colonies (and those with >50% red but <100% red) and dividing by the total number of white, white sectored and less than half red colonies, excluding red colonies from the analysis as they have lost the minichromosome before plating. Results represent data from multiple (2-4) independent cultures of cells, and $5-10 \times 10^3$ colonies were scored for each strain.

*3. Lagging chromosome analysis.* (*Figure 3b*). Chromosome missegregation frequencies were obtained as previously described (*Mellone et al., 2003*). 220 late anaphase cells were scored for presence of lagging chromosomes for each genotype, using two independent cultures of strains.

*4. Mitotic arrest and release* (*Figure 3c*). *nda3-km311* mutant (*Hiraoka et al., 1984*) in H3-WT and H3-G34R backgrounds was used to synchronize cells in mitotic phase. Cells were grown in YES to $5 \times 10^6$ cells/ml at 32°C, then were filtered to rapidly collect cells and transferred to media at 18°C for 8 hr. Cells were released to warm media at 32°C and aliquots fixed at 15 min intervals in 3.7% paraformaldehyde for tubulin and DAPI staining as described in lagging chromosome analysis method.

## Western and dot blots

*1. Denatured extracts in 2xSB*. Whole cell extracts (WCE) were made as published previously (*Alper et al., 2013*). These were used for *Figures 1c, e*, *5b* and *7c*, *Figure 1—figure supplement 1b,g,h,i*, *Figure 1—figure supplement 2a*, *Figure 6—figure supplement 3b*.

*2. Native WCE and chromatin enrichment (NIB buffer)*. Cell lysates were fractionated using a method modified from (*Ekwall et al., 1997*; *Sinha et al., 2010*). Briefly, cell pellets from 50 ml cultures in YES were washed with NIB [0.25 M sucrose, 60 mM KCl, 15 mM NaCl, 5 mM MgCl₂, 1 mM CaCl₂, 2.5 mM spermidine, 20 mM HEPES, 10 mM N-butyric acid, 0.8% Triton X-100 (w/v) with 1:1000 protease inhibitors (Sigma (P8215)) and 1 mM phenylmethylsulfonyl fluoride (PMSF)] and resuspended in 1 ml NIB. Bead beating was performed for 3 min in presence of 500 ul glass beads (Sigma, G8772). After removal of beads, 500 ul of lysate was taken as 'total' extract. The rest was centrifuged at 13,000 rpm for 15 min at 4°C. Supernatant was saved as 'soluble' fraction. Pellet was washed three times in 1 ml NIB buffer, and resuspended in 500 ul NIB buffer as 'chromatin enriched' fraction. Samples were heated with sample buffer prior to PAGE and western. This extract type was used for *Figure 1—figure supplement 1c*.

*3. Denatured extracts in Urea* (*Figure 5c*) Previously published protocols were used for Chk1-HA western blot (*O'Connell et al., 1997*; *Bass et al., 2012*). 25 ml cultures of strains were

treated ±0.05% MMS for 1 hr. MMS was inactivated with freshly made 5% sodium thiosulphate. Cells were disrupted with glass beads (Sigma, G8772) using a bead beater and extracted into urea lysis buffer (8 M urea, 100 mM NaH$_2$PO$_4$, 10 mM Tris pH 8, 60 mM $\beta$-glycerophosphate, 1% Triton X-100 with protease inhibitor cocktail (Sigma P8215) and 1 mM PMSF). The extract was cleared by centrifugation at 13,000 rpm for 10 min, and the supernatant was boiled in SDS sample buffer.

*4. Antibodies used.* **Anti H3** (active motif 39163, lot no. 26311003). **Anti-H4** (millipore 05–858 lot no. 2020541). **Anti-HA** (Roche 12CA5). **Anti-GFP** (Abcam ab290), **Anti-FLAG M2** monoclonal (Sigma F1804), **anti-tubulin** (TAT1 kind gift from Keith Gull)(*Woods et al., 1989*). **Anti-H3K36me2**: Abcam ab9049, **Anti-H3K36me3**: Abcam ab9050, Cell signaling technology 4909. **Anti-Rad51** (Bioacademia 63001). **Anti $\gamma$H2A** (Abcam ab15083, lot no. GR284225-2). **Anti H2A** (active motif 39235, lot no.03108001).

*Western blot quantification* used Image Studio Lite from LiCor. Equal sized squares were drawn using the software, which automatically assigns signal intensity for each band after subtracting background signal. The signal intensities were used to quantify the images, relative to signal intensity for total H3 bands.

5. Peptide sequences used for assessment of K36 methyl Abs (*Figure 1*, and *Figure 1—figure supplement 1*) are listed in *Supplementary file 1*. Dot blots used a 10-fold serial dilution series with 2 ul spots of 1 mM, 0.1 mM, 0.01 mM and 0.001 mM peptides spotted on activated PVDF 0.2 micron membrane. Spots were air dried for 1.5 hr, blocked in 5% BSA in PBST at RT for 1 hr, incubated with primary Ab for 1 hr at RT, washed with PBST, incubated with HRP-conjugated anti-rabbit secondary Ab for 30 min, washed with PBST and then developed with enhanced chemiluminescence and images captured by LiCor imaging. Anti-H3 K36 methylation Abs are listed above and peptide loading was verified by ponceau staining.

## Chromatin immunoprecipitation

ChIP assays (*Figure 1d*, *Figure 2—figure supplement 1b,c*) were performed as described previously (*Alper et al., 2013*). Antibodies for H3K9me2: (Abcam ab1220) and Swi6: (Thermo Scientific PA 1–497). Real time PCR was used to quantify enrichment at heterochromatic loci relative to *adh1$^+$*. Set2-3xFLAG ChIP used Anti-FLAG: (Sigma F1804) and monitored Set2 association with *act1$^+$* or *clr4$^+$* loci as a ratio of signal from input DNA. For Cnp1 ChIPs (*Figure 2—figure supplement 1d*), 10 ul anti-Cnp1 antiserum (*Kniola et al., 2001*) was used per ChIP. Crosslinks were reversed by heating at 100°C for 12 min in the presence of 100 ul Chelex-100 resin (10% slurry in dH$_2$O; BioRad), followed by treatment with Proteinase K (2.5 mg/ml) 30 min at 55°C with shaking and heat-inactivation of Proteinase K (10 min, 100°C). DNA was recovered and used in qPCR analysis. Primer sequences used for q-PCR are listed in *Supplementary file 5*.

## Transcript analysis

*RNA seq studies (Figure 2):* Hot phenol extraction was used to prepare the RNA (*Leeds et al., 1991*). Cultures were grown overnight in YES at 25°C to a density of $2.5 \times 10^6$ cells/ml in a 25 mL culture. The cells were pelleted by centrifugation and washed in DEPC H$_2$O. The pellet was resuspended in 750 ul TES Buffer [50 mM Tris-HCl pH 7.5, 10 mM EDTA, 100 mM NaCl, 0.5% SDS made in DEPC H$_2$O] along with an equal volume of 5:1 phenol:chloroform pH 4.7 and incubated at 65°C for one hour with vortexing for 10 s every 10 min. The samples were then cooled on ice and centrifuged for five minutes at 13,000 RPM. The aqueous phase was transferred to a 2 mL phase lock tube and an additional phenol:chloroform extraction was performed. After centrifugation, the aqueous phase was transferred to a new tube and an equal volume of chloroform was added. To precipitate the RNA, the aqueous phase was transferred to a 2 mL microcentrifuge tube and three volumes of ice cold ethanol and 3M NaOAc pH 5.2 were added and the samples were kept at −20°C to precipitate the RNA. The next day, samples were centrifuged at 14,000 RPM and 4°C for 15 min. The pellet was washed with ice cold 70% ethanol and air dried for 30 min. A Turbo DNAse (Ambion) reaction was set up with 100 ug of RNA in a 150 ul reaction containing 5 ul of Turbo DNAse. The reaction was incubated at 37°C for 30 min and another 5 ul of Turbo DNAse was added and incubated for an additional 30 min prior to the removal of the DNAse using 50 ul of inactivation beads. An RNeasy Mini kit (Sigma) was used to further clean up and concentrate the RNA which was eluted in a final volume of 30 ul of DEPC H$_2$O.

RNA-seq was performed by the St. Jude Hartwell Center. RNA was quantified using a Quant-iT assay (Life Technology). The quality was checked by 2100 Bioanalyzer RNA 6000 Nano assay (Agilent) or LabChip RNA Pico Sensitivity assay (PerkElmer) before library generation. Libraries were prepared from 2 ug of RNA. Ribosomal RNA was removed from the samples using Ribo-Zero Gold rRNA Removal Kit (Yeast) following manufacturer instructions (Illumina). Libraries were prepared using the TruSeq Strand Total RNA Library Prep Kit, beginning at Elution 2 – Fragment – Prime step immediately preceding cDNA synthesis according to the manufacturer instructions (Illumina) with the following modifications; the 94 C Elution 2 – Fragment – Prime incubation was reduced to five minutes and the PCR was reduced to 11 cycles. Libraries were quantified using the Quant-iT Pico-Green dsDNA assay (Life Technologies) or Kapa Library Quantification kit (Kapa Biosystems). One hundred cycle paired end sequencing was performed on an Illumina HiSeq 2500. Three biological replicates were used for each strain analyzed. The total RNA was sequenced using stranded protocol with $2 \times 100$ bp setting. The paired end reads were mapped to *S. pombe* (ftp://ftp.ebi.ac.uk/pub/databases/pombase/pombe/Chromosome_Dumps/Schizosaccharomyces_pombe.ASM294v2.30.dna.genome.fa) genome using STAR. Reads counts for each gene were counted using HTSEQ. Raw counts were TMM and quantile normalized and differentially expressed genes were analyzed using limma/Bioconductor. LogFC was produced from the limma/voom packages (*Law et al., 2014*) in R Bioconductor. The log read count was fit to a linear model of the mean-variance trend using the voom package in R and differentially tested with limma producing the logFC ratios, which were plotted by chromosome location in *Figure 2c* and tabulated in *Supplementary file 3*.

RNA-seq data has been deposited at GEO. Accession no. GSE96842.

Antisense gene transcription data was analyzed using LogFC of 1.5 and $p<0.05$ for triplicate samples, and is tabulated in *Supplementary file 3*.

*Real-time Q-PCR* was performed on random primed cDNA generated from two independent RNA preps for each strain as previously described (*Debeauchamp et al., 2008*; *Partridge et al., 2007*) using primers (*Supplementary file 5*) that had been tested for linear amplification parameters, and working within the Ct range of linear amplification and using an Eppendorf Mastercycler RealPlex$^2$ machine. Transcript levels were normalized to *adh1$^+$* transcripts.

## Preparation of cells for microscopy

*1. DAPI staining for cell imaging (Figure 5e)*. Cells were grown in PMG complete media ± 11 mM HU for 4 hr at 30°C. Cell were fixed in 3.7% paraformaldehyde and stained with DAPI as published previously (*Bass et al., 2012*).

*2. Monitoring markers of DNA damage (Figure 6g and Figure 6—figure supplement 3a,c)*. Cells were grown in YES ±0.05% MMS for 4 hr at 30°C. Cells expressing Rad52–YFP (Rad22-YFP) or Rad11–GFP (RPA) in H3-WT and H3-G34R backgrounds were fixed with 70% ethanol or with 3% formaldehyde. For co-localization, H3-WT and H3-G34R cells co-expressing Rad52–RFP and Rad11-GFP were fixed with 0.5% formaldehyde. Rad51 (Rhp51) was detected by indirect immunofluorescence microscopy as described previously (*Bass et al., 2012*) using cells fixed with 3% paraformaldehyde. Anti-Rhp51 antibody (Thermo scientific Pierce TM PA1-4968) was used at a 1:400 dilution, and detected with FITC-coupled anti-rabbit IgG antibodies at a 1:100 dilution. S129 phosphorylated histone H2A (γH2A) was detected with a phosphorylation-specific antibody (Abcam ab15083) at 1:100 dilution using cells fixed in 1.6% formaldehyde, 0.2% glutaraldehyde. Two independent experiments were performed for each analysis and 300 cells were counted for each genotype and condition.

*3. RPA flare experiment (Figure 5d)*. Cells expressing Rad11–GFP (RPA) were cultured in PMG complete medium to $5 \times 10^6$ cells/ml at 30°C (asynch sample), or were treated with 11 mM HU for 4 hr. Cells were washed by filtration, resuspended in fresh medium and grown at 30°C for 6 hr (6 hr release sample). Cells were fixed with 70% ethanol prior to imaging.

*4. RPA localization during mitosis (Figure 6—figure supplement 3d)*. *cdc10-m17* cells expressing Rad11-GFP (RPA) were cultured in YES medium at 25°C, shifted to the restrictive temperature of 36°C for 4 hr and then released to 25°C. Cells were collected at 30 min intervals, fixed with 3% formaldehyde, and a portion were processed for IF with anti-tubulin antibodies to determine the best time point for capture of late anaphase cells. The remaining cells were DAPI stained and imaged for DAPI and GFP fluorescence.

## Flow cytometry

(*Figure 1—figure supplement 2*, *Figure 6—figure supplement 2*). Cells collected at the indicated time points were fixed in ice-cold 70% ethanol. Cells were washed twice in 20 mM EDTA and incubated with 100 ug/mL RNase A (Amresco E866) overnight at 36C. Prior to analysis, 2 uM of Sytox Green DNA stain (Invitrogen S34860) was added to each sample containing 20 mM EDTA. Flow cytometry was performed on a BD LSR Fortessa cytometer.

## Serial dilution analyses

(*Figure 1—figure supplements 1d*, *4a, c–e* and *6c–e*). Five-fold serial dilution assays were performed using exponentially growing cells and were spotted on agar plates with $1.2 \times 10^4$ cells in the First spot. Plates were incubated at indicated temperatures. For chronic exposure assay, cells were grown to a density of $5 \times 10^6$ cells/ml. A fivefold serial dilution was spotted onto PMG complete agar plates ± HU, on YES agar plates with DMSO or DMSO and CPT, and on YES agar plates ± MMS. Plates were photographed after 4–5 days incubation at 30°C (CPT and MMS), or 6–7 days at 25°C (HU). All experiments were repeated at least twice.

## γIR treatment

(*Figure 4b*). Cells at a density of $5 \times 10^6$ cells/ml were irradiated using a Cobalt source, and 100 ul samples were taken following increments of 200 Gy exposure (*Pai et al., 2014*). Cells were plated on 6 YES plates for each assay condition, and colonies scored following incubation at 32°C for 4 days. The experiment was performed twice to obtain average viability of treated versus untreated cells, and error bars represent the SEM.

## Acute exposure to chemical agents

Protocols were adapted from a previously published protocol (*Sheedy et al., 2005*).

*HU synchronization and release (Figure 7a)*: cells were cultured in PMG complete medium to $5 \times 10^6$ cells/ml at 30°C and were treated with 11 mM HU for 4 hr. Cells were washed by filtration, resuspended in fresh medium and grown at 30°C. Samples were taken at appropriate intervals and counted three times. The experiment was repeated twice and average cell numbers were plotted ± SEM.

*HU synchronization and release of orp1-4 strains (Figure 6b)*: Cells were grown at $25^{°C}$ in PMG complete medium and treated ±11 mM HU for 6.5 hr at 25°C. Cells were washed by filtration, resuspended in fresh medium and incubated at 25 or 36°C for 4 hr before aliquots were plated and incubated at 25°C. Duplicate experiments were performed and data represents mean ± SEM.

*HU synchronization and EdU labeling (Figure 7b)*: This experiment was performed using strains modified to incorporate nucleotide analogs (*Hodson et al., 2003a2003*), and according to the procedure detailed in (*Sabatinos and Forsburg, 2015*). Cells were blocked in 11 mM HU for 4 hr, washed by filtration, resuspended in fresh media containing 10 mM EdU and collected at 15 or 30 min intervals over a 4 hr timecourse, and fixed with 70% ethanol. Controls included incorporating strains that were not exposed to EdU, and a non-incorporating strain that was exposed to EdU. EdU was labeled by use of a ClickIT Alexa Fluor 488 kit (Thermoscientific), and data was collected by FACS analysis on FITC channel. Gating was on cells with fully incorporated EdU. Assay was performed twice, and each sample was read three times. Data from one experiment is shown, with values representing means of triplicate reads with background from control samples subtracted. Error bars represent SD.

## HR assay

(*Figure 6f*, *Figure 6—figure supplement 1a,b,c*): We transformed *leu1-32* mutant cells (that bear a single nucleotide mutation in *leu1* that renders cells auxotrophic for leucine) with a 576 bp fragment of wild type *leu1+* and scored *leu1+* transformants that can only arise by homologous recombination of the wild type *leu1* fragment into the mutant allele. The rates of HR were normalized by calculating transformation efficiencies of the different strains using *leu1+* plasmid DNA. Sequence analysis of ~40 independent *leu1+* recombinants from different genetic backgrounds confirmed that repair always occurred at the *leu1* chromosomal locus.

## Statistical analyses

Non-parametric statistical methods were applied. To compare two independent groups, Wilcoxon-Mann-Whitney (WMW) exact tests were used. Sign test, a non-parametric version of the one sample t-test, was used for cases when data were normalized.

A significance level of 0.05 was used without adjusting for multiplicity. All p-values reported were two-sided.

## Acknowledgements

We thank K Gull for anti-Tat1 Ab, and B Strahl, S Forsburg, T Humphrey, Y Hiraoka, and J Kanoh for gifts of strains and plasmids, and A Carr, S Forsburg, T Humphrey and Junmin Peng for helpful discussions. We thank St. Jude Hartwell Center staff for RNA-seq library preparation, DNA sequencing and peptide synthesis, J Riggs (ARC) for help with the cesium source, S Perry from the Flow Cytometry core, and J Peters and V Frohlich of the cellular imaging core for help and guidance.

## Additional information

### Funding

| Funder | Grant reference number | Author |
|---|---|---|
| St. Baldrick's Foundation | Research grant with generous support from the Henry Cermak fund for pediatric cancer research | Rajesh K Yadav<br>Janet F Partridge |
| National Cancer Institute | CCSG 2 P30 CA21765 | Rajesh K Yadav<br>Carolyn M Jablonowski<br>Alfonso G Fernandez<br>Brandon R Lowe<br>David Finkelstein<br>Jie Huang<br>Arzu Onar-Thomas<br>Janet F Partridge |
| National Institutes of Health | T32 CA078207 | Kevin J Barnum |
| National Institute of General Medical Sciences | GM087326 | Kevin J Barnum<br>Matthew J O'Connell |
| Fox Chase Cancer Center | Board of Associates Fellowship | Ryan A Henry |
| Wellcome | 092076/Z/10/Z | Alison L Pidoux<br>Robin C Allshire |
| National Institute of General Medical Sciences | GM102503 | Andrew J Andrews |
| Wellcome | 095021/Z/10/Z | Robin C Allshire |
| American Lebanese Syrian Associated Charities | | Rajesh K Yadav<br>Carolyn M Jablonowski<br>Alfonso G Fernandez<br>Brandon R Lowe<br>David Finkelstein<br>Jie Huang<br>Arzu Onar-Thomas<br>Janet F Partridge |

The funders had no role in study design, data collection and interpretation, or the decision to submit the work for publication.

### Author contributions

RKY, CMJ, Investigation, Writing—review and editing; AGF, BRL, DF, KJB, ALP, Y-MK, Investigation; RAH, Investigation, Methodology; JH, Formal analysis; MJO'C, AJA, Resources, Supervision, Investigation, Writing—review and editing; AO-T, RCA, Formal analysis, Supervision; JFP,

Conceptualization, Investigation, Writing - original draft preparation, Writing - review and editing, Supervision, Project administration, Funding acquisition

**Author ORCIDs**
Janet F Partridge, http://orcid.org/0000-0003-1102-6305

## Additional files

### Supplementary files

• Supplementary file 1. Peptides used for antibody characterization and mass spectrometry calibration.

• Supplementary file 2. Detection parameters of unique tryptic peptides from *S. pombe* H3.

• Supplementary file 3. Gene expression changes in H3-G34R and *set2Δ* relative to H3-WT.

• Supplementary file 4. *S. pombe* strains.

• Supplementary file 5. Primers used for real time PCR analyses

### Major datasets

The following dataset was generated:

| Author(s) | Year | Dataset title | Dataset URL | Database, license, and accessibility information |
|---|---|---|---|---|
| Yadav RK, Jablonowski CM, Fernandez AG, Lowe BR, Henry RA, Finkelstein D, Barnum KJ, Pidoux AL, Kuo YM, Huang J, O'Connell MJ, Andrews AJ, Onar-Thomas A, Allshire RC, Partridge JF | 2017 | Histone H3-G34R mutation causes replicative stress, defective homologous recombination and genomic instability in Fission Yeast. | https://www.ncbi.nlm.nih.gov/geo/query/acc.cgi?acc=GSE96842 | Publicly available at the NCBI Gene Expression Omnibus (accession no: GSE96842) |

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
