## [Decision Letter]

Thank you for submitting your article "Histone H3-G34R mutation causes replication stress, homologous recombination defects and genomic instability in *S. pombe*" for consideration by *eLife*. Your article has been favorably evaluated by Jessica Tyler (Senior Editor) and three reviewers, one of whom is a member of our Board of Reviewing Editors. The following individual involved in review of your submission has agreed to reveal her identity: Julia Cooper (Reviewer #2).

All three reviewers think highly of your manuscript. The reviewers have discussed the reviews with one another and the Reviewing Editor has drafted this decision to help you prepare a revised submission. Below are the comments that the reviewers agree should be addressed. If you have questions, want to discuss the comments, or send a revision plan ahead of time, please let us know.

1) Figure 1—figure supplement 1 - The purpose of this figure is to convince us that the levels of histones H3 and H4 are the same in a wild-type strain as in the strain with only the single H3/H4 locus. While the western looks nice, the authors should present quantification of three or more western blots.

2) Figure 1 and Figure 1—figure supplement 1 – Similar to the comment above, the other westerns in this manuscript should be presented in a quantitative fashion. These two panels, Figure 1 and Figure 1—figure supplement 1, are another important case – the quantification of the H3K36me3 and H3K36me2 levels in the G34R mutant. Why do Figure 1 and Figure 1—figure supplement 1 both show westerns analyzing H3K36me3 levels? What's the point of showing this twice? One representative western accompanied by quantification of multiple westerns would be best.

3) Figure 1 – In this figure, there is no detectable H3K36me3 in the G34 mutant, whereas it's easily seen, although reduced, in Figure 1. Why is there this difference? Is the result variable? Some comment should be included. As for other westerns, quantification is needed for these results. (And please state what EV stands for.)

4) Several studies in both *S. pombe* and *S. cerevisiae* have shown that Set2 functions to repress antisense transcription. Examples of *S. pombe* papers are Nicolas et al. (2007) NSMB 14, 372 and Shim et al. (2012) Embo J. 31, 4375. The most recent paper for *S. cerevisiae* studies is Vankatesh et al. (2016) Nature Comm. As the authors performed strand-specific RNA-seq, they have the data to a set2 mutant and the G34R mutant. This would provide more information regarding the possible transcriptional differences between the two mutants.

5) The Results section about the transcriptional profiling describes the results, but it leaves us hanging. Can the authors state a conclusion at the end of the section? It appears that these results clearly show that the phenotypes caused by G34R are distinct from set2 and therefore are not likely caused by the reduction of H3K36me3. This is emphasized by opposite effects on the transcript levels of some genes (nicely demonstrated in Figure 2). As the authors study a G34R set2 double mutant in later experiments, it would be interesting, although not necessary, to measure *fah1*^+^ and *grt1*^+^ mRNA levels in the G34R set2 double mutant.

6) Previous studies have shown that K36 acetylation and methylation peak at different times during the cell cycle (Pai et al., Nat Comm 2014; Li et al., Cell 2013) and that these events are correlated with DNA repair pathway choice. The consequences of H3-G34R on H3-K36 modification should thus be assessed in the context of cell cycle progression.

7) Please say more about the sub-subtelomeric regions affected differentially in the set2 mutant versus the G34R mutant – are they the Sgo2-associated 'knob' regions? It could be informative to check Sgo2 loading/function, particularly as it plays important roles at centromeres, which are likely sensitive sites for the replication difficulties conferred by G34R. ChIP-seq of phospho-H2A would also be informative, although it seems beyond the scope of the current studies.

8) As one of the key phenotypes of H3-G34R is genomic instability that may be related to DNA replication, characterization of cell cycle progression in this background should be provided. This includes an analysis of the effect of this alteration on the generation time, the cell cycle phase distribution, and the length of S phase (by flow cytometry, for instance). These data would provide valuable information for interpreting the defects observed in these cells. For instance, NHEJ is normally virtually dispensable in wild-type fission yeast (Ferreira and Cooper, Genes Dev 2004), which have a very short G1. However, as G34R appears to make NHEJ crucial for viability (Figure 6), one possibility is that cells are spending more time in G1, making NHEJ an important repair pathway. The authors can use their *nda3-km311* strain to synchronize cells for flow cytometry for cell cycle analysis.

9) The authors state in several places that RPA intensities are unaffected by G34R, but this is not clear from the images – in Figure 6—figure supplement 2, it looks like, while wild type responds to damage with increased RPA intensity/number, this does not happen in G34R. In Figure 5 as well, it looks like there are less RPA in G34R. Please define the quantitation method. It would be useful to look at RPA during mitosis, when it may coat ultrafine bridges left behind from unresolved replication problems. The affected H3 modifications may promote robust resection and RPA loading. This should be re-examined and ideally, a site specific DSB induced so that resection extent could be measured.

10) A role for methylation of H3-K36 by Set2 has been identified in suppressing homologous recombination in the fission yeast (Pai et al., Nat Comm 2014). This contrasts with the results presented in Figure 6—figure supplement 1, which show a reduction in HR efficiency in *set2Δ*. The authors do not address this difference, which is significant for the interpretation of the results for H3-G34R.

11) The HR assay used in this work involves the integration of a linear PCR fragment containing the *leu1* gene. Given that the H3-G34R background displays increased replications stress and that cells can use both NHEJ and HR pathways to repair double-stranded DNA breaks, it is important to demonstrate that 1) the majority of these events occur at the chromosomal locus and/or 2) an NHEJ mutant such as *ku70Δ* does not alter the HR efficiency measured in this assay. This is particularly relevant as a previous study using a human cell line has reported that H3-K36 dimethylation enhances the repair of double-stranded DNA breaks by NHEJ (Fnu et al., PNAS 2011).

12) It appears that there may be different sets of phenotypes induced by H3-G34R: 1) those related to H3-K36 modification/Set2, which may include the effect on genome stability, and 2) and those that the authors state are unlikely to be due to modulation of H3-K36, such as the defect in chromosome segregation. The possibility that H3-G34R affects chromosome stability and repair choice by mechanisms other than through H3-K36 is interesting, and while this is discussed, the reader would benefit from a more clear presentation of these potentially distinct functions.

13) In addition to the westerns, the number of replicates for some experiments is not always clear. Please provide that information and the relevant statistical analysis.

14) The protocol for *orp1-4* utilization in Figure 6 is confusing compared with text and Methods.

---

## [Author Response]

*[…] 1) Figure 1—figure supplement 1 - The purpose of this figure is to convince us that the levels of histones H3 and H4 are the same in a wild-type strain as in the strain with only the single H3/H4 locus. While the western looks nice, the authors should present quantification of three or more western blots.*

Thank you for this suggestion. As requested, we ran westerns to allow quantitation from 4 biological replicates per strain, and the mean results for single copy H3 and H4 levels relative to triple copy histones, normalized to α-tubulin are displayed, along with error bars reflecting the standard error of the mean (SEM) in Figure 1—figure supplement 1.

*2) Figure 1 and Figure 1—figure supplement 1 – Similar to the comment above, the other westerns in this manuscript should be presented in a quantitative fashion. These two panels, Figure 1 and Figure 1—figure supplement 1, are another important case – the quantification of the H3K36me3 and H3K36me2 levels in the G34R mutant. Why do Figure 1 and Figure 1—figure supplement 1 both show westerns analyzing H3K36me3 levels? What's the point of showing this twice? One representative western accompanied by quantification of multiple westerns would be best.*

The anti-H3K36me3 Ab used in Figure—figure supplement 1F and G is a different antibody to that used in Figure 1, and the experiments in the supplement were included to demonstrate reproducibility of the data with different reagents. As suggested, we ran replicate westerns for K36me2 and K36me3 (using the Abs used in Figure 1/C) including 2 biological replicates for H3-WT and H3-G34R strains for quantitation. The mean results for K36me2 or 3 relative to total H3 are displayed on the right-hand panel of Figure 1.

*3) Figure 1 – In this figure, there is no detectable H3K36me3 in the G34 mutant, whereas it's easily seen, although reduced, in Figure 1. Why is there this difference? Is the result variable? Some comment should be included. As for other westerns, quantification is needed for these results. (And please state what EV stands for.)*

Thanks for pointing this out. The result is not variable – we had just used a shorter exposure. We have replaced Figure 1 and provided quantitation from 3 western blots generated using 2 biological replicates for each strain. EV stands for empty vector – we have included this in the figure legend.

*4) Several studies in both S. pombe and S. cerevisiae have shown that Set2 functions to repress antisense transcription. Examples of S. pombe papers are Nicolas et al. (2007) NSMB 14, 372 and Shim et al. (2012) Embo J. 31, 4375. The most recent paper for S. cerevisiae studies is Vankatesh et al. (2016) Nature Comm. As the authors performed strand-specific RNA-seq, they have the data to a set2 mutant and the G34R mutant. This would provide more information regarding the possible transcriptional differences between the two mutants.*

We assessed antisense transcripts in H3-WT, H3-G34R and set2D, and show in Figure 2 plots for antisense and sense transcripts that are differentially regulated in set2Δ compared to H3-WT (upper panel) and H3-G34R compared to H3-WT (lower panel). Like set2Δ, G34R shows accumulation of antisense transcripts. Lists of differentially regulated antisense transcripts are provided in [Supplementary-material SD3-data] (antisense tab).

*5) The Results section about the transcriptional profiling describes the results, but it leaves us hanging. Can the authors state a conclusion at the end of the section? It appears that these results clearly show that the phenotypes caused by G34R are distinct from set2 and therefore are not likely caused by the reduction of H3K36me3. This is emphasized by opposite effects on the transcript levels of some genes (nicely demonstrated in Figure 2). As the authors study a G34R set2 double mutant in later experiments, it would be interesting, although not necessary, to measure fah1^+^ and grt1^+^ mRNA levels in the G34R set2 double mutant.*

Interpretation of the transcriptional profiling results is complex. As stated, there is some overlap with Set2-mediated control which is now reinforced by the data on accumulation of antisense transcripts (Figure 2), but also differences as evidenced by opposite regulation of transcription within sub-subtelomeric domains. The additional experiments suggested by reviewers have now helped clarify the situation- thank you. As suggested we have included analysis of *fah1^+^* and *grt1^+^* mRNA levels in *set2Δ* H3-G34R double mutants (Figure 2), and find that the results mimic data obtained from *set2Δ*. From these additional analyses we conclude that suppression of antisense transcripts appears to be linked to trimethylation of K36 (reduced or lost in H3-G34R and *set2Δ*), and that repression of sub-subtelomeric domains may be linked to dimethylation of K36 in H3-G34R, as it is abolished in *set2Δ* or compound *set2Δ* H3-G34R mutants.

*6) Previous studies have shown that K36 acetylation and methylation peak at different times during the cell cycle (Pai et al., Nat Comm 2014; Li et al., Cell 2013) and that these events are correlated with DNA repair pathway choice. The consequences of H3-G34R on H3-K36 modification should thus be assessed in the context of cell cycle progression.*

We harvested H3-WT and H3-G34R mutant cells arrested by *cdc10* (G1), HU (S phase), *nda3* (metaphase) and *cdc25-22* (G2) mutation / treatment and monitored K36me2 and me3 (Figure 1—figure supplement 2). Quantification of 3 westerns from two biological replicates for each condition revealed that at all cell cycle points, K36me3 was reduced in H3-G34R mutants, and that H3K36me2 was elevated in H3-G34R. Interestingly, G1 arrested (*cdc10*) cells showed increased K36me3 for both H3-WT and H3-G34R cells, suggestive that the cell cycle regulation of Set2 function is maintained in G34R cells, even though G34R mutation generally diminishes Set2 function. We were not able to easily measure K36Ac as all tested antibodies are unable to recognize K36Ac on the G34R tail so this was not assessed through the cell cycle.

*7) Please say more about the sub-subtelomeric regions affected differentially in the set2 mutant versus the G34R mutant – are they the Sgo2-associated 'knob' regions? It could be informative to check Sgo2 loading/function, particularly as it plays important roles at centromeres, which are likely sensitive sites for the replication difficulties conferred by K34R. ChIP-seq of phospho-H2A would also be informative, although it seems beyond the scope of the current studies.*

Yes, the sub-subtelomeric regions are the Sgo2-associated ‘knob’ regions. We plan to include Sgo2 ChIPs in a follow up manuscript comparing H3-G34R and V mutant phenotypes.

*8) As one of the key phenotypes of H3-G34R is genomic instability that may be related to DNA replication, characterization of cell cycle progression in this background should be provided. This includes an analysis of the effect of this alteration on the generation time, the cell cycle phase distribution, and the length of S phase (by flow cytometry, for instance). These data would provide valuable information for interpreting the defects observed in these cells. For instance, NHEJ is normally virtually dispensable in wild-type fission yeast (Ferreira and Cooper, Genes Dev 2004), which have a very short G1. However, as G34R appears to make NHEJ crucial for viability (Figure 6), one possibility is that cells are spending more time in G1, making NHEJ an important repair pathway. The authors can use their nda3-km311 strain to synchronize cells for flow cytometry for cell cycle analysis.*

We monitored generation time during exponential growth and found that H3-WT have a doubling time of 180 min (+/- 10 min), H3-G34R 201 min (+/- 4 min), and *set2Δ* 194 min (+/- 8 min) in rich media at 32^o^C. We also performed cell cycle synchronization and release using *cdc10* (G1 arrest) and *nda3* (mitotic) blocks and used flow cytometry to monitor cell cycle progression. As shown in Figure 6—figure supplement 2, *cdc10* released H3-G34R cells exhibit delay in completion of S phase. We were not able to decipher if G1 was extended as fission yeast do not undergo cytokinesis until G1, which hampers interpretation of FACS.

*9) The authors state in several places that RPA intensities are unaffected by K34R, but this is not clear from the images – in Figure 6—figure supplement 2, it looks like, while wild type responds to damage with increased RPA intensity/number, this does not happen in G34R. In Figure 5 as well, it looks like there are less RPA in G34R. Please define the quantitation method. It would be useful to look at RPA during mitosis, when it may coat ultrafine bridges left behind from unresolved replication problems. The affected H3 modifications may promote robust resection and RPA loading. This should be re-examined and ideally, a site specific DSB induced so that resection extent could be measured.*

The data presented was not the best choice of images. We have re-assessed RPA foci in cells using more robust fixation (3% formaldehyde) and have replaced images in what is now Figure 6—figure supplement 3. For the initial submission we had manually counted several hundred cells for each genotype for foci and repeated the complete experiment 2-3 times. The images obtained with formaldehyde fixation better represent the results. We attempted to use automated counting of cells bearing foci, but this was complicated since cells bear distinct numbers of foci and of different sizes, but limited data obtained through this approach supported that there was no difference in RPA foci intensity between MMS treated strains. In the absence of a system to quickly assess RPA loading dynamics at a defined break by ChIP, we can just conclude that there is no defect in initial detection of damage in H3-G34R cells as defined by intact checkpoint responses and γH2A signaling. Instead, there appears to be a defect at a later stage of HR damage repair prior to Rad51 and 52 loading.

As suggested, we monitored RPA-GFP localization in H3-G34R cells during mitosis, using cells synchronized by *cdc10* block and release, and found evidence for RPA fluorescence linking DAPI stained masses in cells enriched for anaphase. This is an exciting result, as it reinforces that G34R confers replicative difficulties that may contribute to chromosome segregation defects seen in these strains. This data is included in Figure 6—figure supplement 3. We were unable to quantify numbers of late anaphase cells displaying this phenotype since we could not co-stain with tubulin antibodies and still see the faint GFP signal on anaphase bridges due to bleed through issues. However, cells displaying these anaphase bridges were relatively easy to detect, suggestive that they occur at relatively high frequency.

*10) A role for methylation of H3-K36 by Set2 has been identified in suppressing homologous recombination in the fission yeast (Pai et al., Nat Comm 2014). This contrasts with the results presented in Figure 6—figure supplement 1, which show a reduction in HR efficiency in set2Δ. The authors do not address this difference, which is significant for the interpretation of the results for H3-G34R.*

We have been concerned by this difference, which was the stimulus for assessing HR in 3xH3/H4 *set2Δ* strains (Figure 6—figure supplement 1). Pai et al. 2014 assessed frequencies of gene conversion using an engineered non-essential minichromosome with an HO-break site. The authors monitor 4 different markers for loss, and interpret combinations of marker loss as NHEJ-repaired, Gene conversion, loss of heterozygosity and loss of the chromosome. However, due to the nature of the assay, there are many possible causes of apparent marker loss which may cloud interpretation. Our simple and direct approach to measure HR relies on a site-specific gain of marker function at a defined locus. We note that we have just assessed HR at one site, and it is quite feasible that distinct outcomes may be obtained by assessing HR at different genomic regions. We sought to determine if the difference in data was caused by the different assays or possibly by strain differences between the *set2Δ* backgrounds. Using *set2Δ* strains from Tim Humphrey’s lab that were used in Pai et al., we monitored HR using our assay, and found that HR activity was suppressed similar to the results obtained with our *set2Δ* 3xH3/H4 strains. This data is included in Figure 6—figure supplement 1. We conclude that the difference in interpretation lies with the assays used.

*11) The HR assay used in this work involves the integration of a linear PCR fragment containing the leu1 gene. Given that the H3-G34R background displays increased replications stress and that cells can use both NHEJ and HR pathways to repair double-stranded DNA breaks, it is important to demonstrate that 1) the majority of these events occur at the chromosomal locus and/or 2) an NHEJ mutant such as ku70Δ does not alter the HR efficiency measured in this assay. This is particularly relevant as a previous study using a human cell line has reported that H3-K36 dimethylation enhances the repair of double-stranded DNA breaks by NHEJ (Fnu et al., PNAS 2011).*

The HR assay was designed so that only recombinants at the correct locus would yield *leu1+* colonies, since only a 576 bp PCR fragment of *leu1* was used. We have confirmed this by sequencing *leu1*+ in ~ 40 “repaired” strains from different genetic backgrounds and all were correctly targeted. We were interested to see the effect of loss of NHEJ function in our assay, so tested strains deficient in Ku70. Perhaps as expected, these strains showed a large increase in HR frequency in our assay (Figure 6—figure supplement 1), as HR is now the sole method of DNA break repair. Discussion of these results has been incorporated within the text.

*12) It appears that there may be different sets of phenotypes induced by H3-G34R: 1) those related to H3-K36 modification/Set2, which may include the effect on genome stability, and 2) and those that the authors state are unlikely to be due to modulation of H3-K36, such as the defect in chromosome segregation. The possibility that H3-G34R affects chromosome stability and repair choice by mechanisms other than through H3-K36 is interesting, and while this is discussed, the reader would benefit from a more clear presentation of these potentially distinct functions.*

We agree that the phenotype of H3-G34R mutants is complex and very interesting. We hope that the additional analyses included in this revision have helped to define which aspects of G34R function are possibly linked to modification of H3K36, such as a requirement for trimethylation of K36 for suppression of antisense gene transcription. We are very excited to see the localization of RPA to anaphase bridges in mitotic cells, as this reinforces that replicative defects likely contribute to the chromosome segregation defects in H3-G34R mutants. However, in the absence of separation of function mutants, it is very difficult to determine whether retention or accumulation of K36 dimethylation in H3-G34R cells is linked to the replicative defects, or whether reduction in K36 Ac is a contributory factor. What is clear is that further analysis is required to fully understand the behavior of H3-G34R, but we believe that our work represents a highly significant advance in characterization of this mutant.

*13) In addition to the westerns, the number of replicates for some experiments is not always clear. Please provide that information and the relevant statistical analysis.*

We have added this information to the manuscript.

*14) The protocol for orp1-4 utilization in Figure 6 is confusing compared with text and Methods.*

Thanks for catching this. We have modified the text of the figure legend and the cartoon in the figure.